# On the Expressive Power of Mixture-of-Experts for Structured Complex Tasks

**Mingze Wang**
School of Mathematical Sciences, Peking University, Beijing, China
mingzewang@stu.pku.edu.cn

**Weinan E**
Center for Machine Learning Research and School of Mathematical Sciences, Peking University, Beijing, China
AI for Science Institute, Beijing, China
weinan@math.pku.edu.cn

## Abstract

Mixture-of-experts networks (MoEs) have demonstrated remarkable efficiency in modern deep learning. Despite their empirical success, the theoretical foundations underlying their ability to model complex tasks remain poorly understood. In this work, we conduct a systematic study of the expressive power of MoEs in modeling complex tasks with two common structural priors: low-dimensionality and sparsity. For shallow MoEs, we prove that they can efficiently approximate functions supported on low-dimensional manifolds, overcoming the curse of dimensionality. For deep MoEs, we show that $\mathcal{O}(L)$-layer MoEs with $E$ experts per layer can approximate piecewise functions comprising $E^L$ pieces with compositional sparsity, i.e., they can exhibit an exponential number of structured tasks. Our analysis reveals the roles of critical architectural components and hyperparameters in MoEs, including the gating mechanism, expert networks, the number of experts, and the number of layers, and offers natural suggestions for MoE variants.

## 1 Introduction

Mixture-of-experts (MoE) models (Jacobs et al., 1991; Jordan and Jacobs, 1994) have recently achieved significant success in deep learning, particularly as a core architectural component of modern large language models (LLMs) (Abdin et al., 2024; Yang et al., 2024b; Liu et al., 2024; Cai et al., 2025). These models have demonstrated strong capabilities across a wide range of complex and diverse tasks, including mathematical reasoning, logical inference, language understanding, and code generation. Despite their empirical success, the theoretical foundations underlying MoEs remain poorly understood, especially in their capacity to efficiently model complex tasks.

In both machine learning and applied mathematics, it is widely recognized that although real-world tasks may appear complex, they often exhibit latent structures. Two prominent structural priors are: (1) *low-dimensional structure*: high-dimensional data typically lies on a manifold of much lower intrinsic dimension; (2) *sparse structure*: meaningful signals tend to admit sparse representations in suitable bases or dictionaries. These structural priors have motivated numerous influential algorithms, including dimensionality reduction (Tenenbaum et al., 2000), sparse regression via Lasso (Tibshirani, 1996), compressed sensing (Donoho, 2006), and neural network pruning and compression techniques.

In this work, we investigate the expressive power of MoE networks for modeling complex tasks that exhibit either low-dimensional or sparse structure. **Our contributions** are summarized as follows:

39th Conference on Neural Information Processing Systems (NeurIPS 2025).

- **Shallow MoE networks.** We prove that shallow MoE networks can efficiently approximate functions supported on *low-dimensional manifold*. Theoretically, this task reduces to a collection of simpler approximation subproblems localized on low-dimensional subregions, along with an assignment problem that maps each input to the appropriate region. We show that shallow MoE networks naturally implement this procedure, thereby avoiding the curse of dimensionality. Our analysis reveal the complementary roles of the two core components in MoE: expert networks approximate localized subfunctions, while the gating mechanism ensures correct input-to-expert assignment. Additionally, the analysis offers practical suggestions on MoE variants, such as the nonlinear gating, alternating MoE architectures with equivalent expressivity, and low-dimensional expert networks with auto-encoding.

- **Deep MoE networks.** We formalize complex tasks as piecewise functions, and focus on a broad class of structured tasks exhibiting *compositional sparsity*: where each subtask depends on only a small subset of input coordinates, and the overall task is a hierarchial composition of these subtasks. We demonstrates that a depth-$\mathcal{O}(L)$ MoE network with $E$ expert per layers can efficiently approximate piecewise functions with $E^L$ distinct pieces, i.e, it can exhibit an exponential number of structured tasks. Moreover, our analysis elucidates the distinct roles of network depth $L$ (which enables hierarchical composition) and expert count $E$ (which enables subtask specialization).

- **Unified insights.** Our theoretical results reveal that MoE networks can effectively discover the underlying structure priors in the complex tasks (such as low-dimensionality or sparsity), and subsequently decompose them into simpler subproblems, each solved by specialized experts.

## 2 Related Works

**Theoretical understanding of MoE**. Chen et al. (2022) analyzed the training dynamics of shallow MoE networks with softmax gating on clustered datasets, emphasizing the importance of expert nonlinearity and data structure. (Baykal et al., 2022) showed that sparsely activated networks can achieve approximation performance comparable to dense networks, and offered a computationally efficient alternative. Dikkala et al. (2023) examined the impact of learnable routing mechanisms in MoEs, establishing their benefits. Li et al. (2024) investigated MoE in continual learning, using overparameterized linear regression to show their adaptability across tasks. A comprehensive survey of recent theoretical advances is presented in Mu and Lin (2025). In contrast to these prior works, we focus on the expressive power of both shallow and deep MoE networks for broad classes of structured functions.

**Low-dimensional structure.** The *manifold hypothesis* posits that high-dimensional data in real world (e.g., images, speech, and text) typically lies on a manifold of much lower intrinsic dimensionality than the ambient space. This perspective motivates various algorithmic approaches: *(i) Dimensionality reduction* techniques (Tenenbaum et al., 2000; Roweis and Saul, 2000; Belkin and Niyogi, 2003), which aim to uncover and utilize such low-dimensional structures. *(ii) Representation learning* methods like Autoencoders and Variational Autoencoders (Hinton and Salakhutdinov, 2006; Kingma et al., 2013), which seek compact and informative representations aligned with low-dimensional manifold.

**Sparse structure.** It is widely believed that meaningful signals often admit sparse representations in appropriate bases or dictionaries. This principle underpins many influential algorithms, such as Lasso (Tibshirani, 1996), Compressed Sensing (Donoho, 2006; Candès and Wakin, 2008), and Sparse Coding (Olshausen and Field, 1996; Elad and Aharon, 2006), which have been widely applied across domains.

From a theoretical standpoint, the prevalence of sparsity and low-dimensionality has inspired recent studies on the expressive power of deep networks under structural assumptions. For example, Mhaskar and Poggio (2016); Poggio (2023) analyzed dense neural networks approximating functions with compositional sparsity, demonstrating how sparsity mitigates the curse of dimensionality (Bellman, 1966; Bach, 2017). Wang et al. (2024) studied the expressivity of Transformer models (Vaswani et al., 2017) for modeling long but sparse memories, showing the model's capacity to overcome the curse of memory. Shaham et al. (2018); Chen et al. (2019) examined the approximation power of dense networks for functions supported on low-dimensional manifolds. In contrast to these works, we

investigates the expressive power of MoE networks in approximating complex functions exhibiting either sparse or low-dimensional structure.

# 3 Preliminaries

**Basic notations.** Let $f : \Omega \to \mathbb{R}$ be a continuous function defined on a compact set $\Omega$. Its $L_\infty$ norm is defined as $\|f\|_{L_\infty} := \sup_{\boldsymbol{x} \in \Omega} |f(\boldsymbol{x})|$. We use standard asymptotic notations $\mathcal{O}(\cdot), \Omega(\cdot), \Theta(\cdot)$ to hide the constants independent of the primary problem size (typically denoted by $m$), and the notations $\tilde{\mathcal{O}}(\cdot), \tilde{\Omega}(\cdot), \tilde{\Theta}(\cdot)$ further hide logarithmic factors. For a positive integer $n$, let $[n] = \{1, \cdots, n\}$. For $a, b \in \mathbb{R}$, define $a \wedge b = \min\{a, b\}$ and $a \vee b = \max\{a, b\}$.

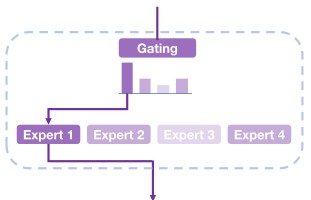

Figure 1: Illustration of an MoE layer.

## 3.1 MoE networks

**MoE components.** An MoE layer consists of two primary components:

- **Expert networks:** A collection of $E$ expert networks, each implemented as a dense feedforward ReLU neural network: $f^{(1)}, \cdots, f^{(E)} : \mathbb{R}^{d_{\text{in}}} \to \mathbb{R}^{d_{\text{out}}}$.
- **Gating network:** A gating function $g : \mathbb{R}^{d_{\text{in}}} \to \mathbb{R}^E$. In most existing MoE models (Fedus et al., 2022; Du et al., 2022; Yang et al., 2024a), $g$ is linear due to its simplicity and empirical effectiveness: $g(\boldsymbol{x}) = \boldsymbol{W}_R \boldsymbol{x}$, where $\boldsymbol{W}_R \in \mathbb{R}^{E \times d_{\text{in}}}$.

**MoE operation.** Given an input $\boldsymbol{x} \in \mathbb{R}^{d_{\text{in}}}$, an MoE layer performs the following operations, as illustrated in Figure 1:

- **Expert selection.** The gating network computes routing scores $g(\boldsymbol{x}) \in \mathbb{R}^E$, and selects the top-$K$ experts with the highest scores:
$$\mathcal{K} := \arg \text{TopK}(g(\boldsymbol{x})),$$
where $\arg \text{TopK}(\boldsymbol{z})$ returns the indices of the $K$ largest entries of $\boldsymbol{z}$.

- **Expert computation and aggregation.** Each selected expert $k \in \mathcal{K}$ computes its output $f^{(k)}(\boldsymbol{x})$. The final output is a weighted combination:
$$\boldsymbol{y} = \sum_{k \in \mathcal{K}} \alpha_k(\boldsymbol{x}) f^{(k)}(\boldsymbol{x}),$$
where the weight are defined via $\alpha_k(\boldsymbol{x}) = \frac{\exp(g_k(\boldsymbol{x}))}{\sum_{j \in \mathcal{K}} \exp(g_j(\boldsymbol{x}))}$.

Notably, only $K$ expert networks are activated per input. Without loss of generality, we focus throughout this paper on **the case $K = 1$**, as the extension to arbitrary $K \leqslant E$ is straightforward.

**Hypothesis class $\mathcal{H}_{l,m}^{L,E}$.** We define $\mathcal{H}_{l,m}^{L,E}$ as the class of depth-$L$ neural networks composed of stacked $L$ MoE layers:
$$h^{(L)} \circ h^{(L-1)} \circ \cdots \circ h^{(1)}, \tag{1}$$
where each $h^{(\ell)}$ is an MoE layer consisting of a linear gating network $g^{(\ell)}$ and $E$ expert networks $f^{(\ell,e)}$ ($e \in [E]$), each being an $l$-layer, $m$-width dense ReLU neural network.

## 3.2 Classical approximation results

**Approximation error notation.** Let $\mathcal{E}_{l,m}^{\text{FFN}}(f)$ denote the $L^\infty$ approximation error of a target function $f : \Omega \to \mathbb{R}$ using $l$-layer, $m$-width dense ReLU neural networks.

$\mathcal{C}^K$ **space.** Let $D, K \in \mathbb{N}$, and $\Omega \subset \mathbb{R}^D$ be compact. The space $\mathcal{C}^K(\Omega)$ consists of all functions $f$ such that
$$\|f\|_{\mathcal{C}^K(\Omega)} = \max_{0 \leqslant \|\boldsymbol{\beta}\|_1 \leqslant K} \|D^{\boldsymbol{\beta}} f\|_{L_\infty(\Omega)} < \infty \tag{2}$$

where $D^{\boldsymbol{\beta}} f$ denotes the partial derivatives of order $\boldsymbol{\beta} = (\beta_1, \cdots, \beta_D) \in \mathbb{Z}_+^D$. The space of **smooth functions** is defined as $\mathcal{C}^\infty(\Omega) = \bigcap_{K \in \mathbb{N}} \mathcal{C}^K(\Omega)$. Additionally, the **smoothness exponent** of $f : \Omega \to \mathbb{R}$ is defined as

$$\kappa(f) := \sup\{K \in \mathbb{N} : f \in \mathcal{C}^K(\Omega)\}. \tag{3}$$

The following result summarizes the classical approximation rate of two-layer ReLU networks for $\mathcal{C}^K$ functions (Mao and Zhou, 2023; Yang and Zhou, 2024):

**Theorem 3.1.** *Let $D, K \in \mathbb{N}$, and $\Omega \subset \mathbb{R}^D$ be compact. For any $f \in \mathcal{C}^K(\Omega)$ and $m \in \mathbb{N}$, there exits a two-layer ReLU neural network $f_m$ with $m$ hidden neurons such that*

$$\mathcal{E}_{2,m}^{\mathrm{FFN}}(f) \leqslant \|f - f_m\|_{L_\infty(\Omega)} \leqslant \begin{cases} \mathcal{O}\big(m^{-\frac{K}{D}}\big), & \text{if } K < \frac{D+3}{2}, \\ \tilde{\mathcal{O}}\big(m^{-\frac{1}{2}}\big), & \text{otherwise.} \end{cases}$$

When the smoothness of the target function is relatively low, i.e., $K \ll D$, the approximation rate $\mathcal{O}\big(m^{-\frac{K}{D}}\big)$ reveals that two-layer networks suffer from the curse of dimensionality (CoD).

# 4 Theory for Shallow MoE Networks

In this section, we study the efficiency of shallow MoE networks in approximating functions supported on a low-dimensional manifold $\mathcal{M}$.

## 4.1 Manifold in Euclidean Space

Let $\mathcal{M}$ be a $d$-dimensional smooth manifold embedded in $\mathbb{R}^D$. We begin by reviewing several standard definitions.

**Definition 4.1** (Chart and Atlas).

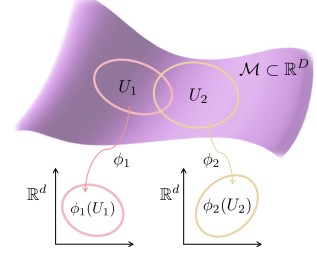

- A *chart* for $\mathcal{M}$ is a pair $(U, \phi)$ such that $U \subset \mathcal{M}$ is open and $\phi : U \to \mathbb{R}^d$, where $\phi$ is a homeomorphism (i.e., bijective, $\phi$ and $\phi^{-1}$ are both continuous). $U$ is called a coordinate neighborhood, and $\phi$ is the associated coordinate map.

Figure 2: A $d$-dimensional manifold $\mathcal{M}$ in $\mathbb{R}^D$.

- An *atlas* of $\mathcal{M}$ is a collection $\{(U_\alpha, \phi_\alpha)\}_{\alpha \in \mathcal{A}}$ of charts such that $\cup_{\alpha \in \mathcal{A}} U_\alpha = \mathcal{M}$.

An atlas $\{(U_\alpha, \phi_\alpha)\}_{\alpha \in \mathcal{A}}$ is called *smooth* if for any overlapping charts $(U_\alpha, \phi_\alpha)$ and $(U_{\alpha'}, \phi_{\alpha'})$, the transition maps $\phi_\alpha \circ \phi_{\alpha'}^{-1}$ and $\phi_{\alpha'} \circ \phi_\alpha^{-1}$ are smooth functions.

**Definition 4.2** (Smooth manifold). The manifold $\mathcal{M}$ is called *smooth* if it has a smooth altas.

We now introduce the partition of unity, which can divide the manifold into regular subregions.

**Definition 4.3** (Partition of unity). Let $\{U_\alpha\}_{\alpha \in \mathcal{A}}$ be an open cover of $\mathcal{M}$. A **partition of unity** of $\mathcal{M}$ w.r.t this cover is a family of nonnegative smooth functions $\rho_\alpha : \mathcal{M} \to [0, 1]$ for $\alpha \in \mathcal{A}$ such that:

- *(i)* for all $\alpha \in \mathcal{A}$, $\rho_\alpha$ has compact support and $\mathrm{supp}(\rho_\alpha) \subset U_\alpha$;

- *(ii)* for every $\boldsymbol{x} \in \mathcal{M}$, only finitely many $\rho_\alpha(\boldsymbol{x})$ are nonzero;

- *(iii)* for all $\boldsymbol{x} \in \mathcal{M}$, $\sum_{\alpha \in \mathcal{A}} \rho_\alpha(\boldsymbol{x}) = 1$.

**Theorem 4.4** (Existence of a partition of unity). *Let $\{U_\alpha\}_{\alpha \in \mathcal{A}}$ be an open cover of a smooth manifold $\mathcal{M}$. Then there exists a partition of unity $\{\rho_\alpha\}_{\alpha \in \mathcal{A}}$ of $\mathcal{M}$ w.r.t. $\{U_\alpha\}_{\alpha \in \mathcal{A}}$.*

We next define the smoothness of a function defined on a manifold.

**Definition 4.5** (Function on the manifold). Let a function $f : \mathcal{M} \to \mathbb{R}$, and $\{(U_\alpha, \phi_\alpha)\}_{\alpha \in \mathcal{A}}$ be a smooth atlas of $\mathcal{M}$. Its smoothness $\kappa(f)$ is defined by $\kappa(f) := \inf_{\alpha \in \mathcal{A}} \kappa(f \circ \phi_\alpha^{-1})$, where the smoothness of each $f \circ \phi_\alpha^{-1}$ is defined as Equation (3).

In this paper, we focus on **smooth compact manifolds**. Due to the compactness of $\mathcal{M}$, its atlas consists of a finite collection of charts, denoted by $\{(U_i, \phi_i)\}_{i \in [E]}$. Additionally, we can let $\phi_i(U_i) \subset [0,1]^d$. By Theorem 4.4, there exists a corresponding partition of unity $\{\rho_i\}_{i \in [E]}$.

Compact smooth manifolds admit an atlas with strong geometric regularity as below, which is detailed in Appendix A.

**Example 4.6** (Highly regular atlas). Let $\mathcal{M}$ be a compact smooth manifold. Then there exists a highly smooth atlas $\{(U_i, \phi_i)\}_{i \in [E]}$ such that each map $\phi_i : U_i \to [0,1]^d$ is a *linear function*. Thus, each $\phi_i$ satisfies $\kappa(\phi_i) = \infty$ (when viewed as a function in $\mathbb{R}^D$).

Motivated by this example, we define a broad class of regular atlas:

**Definition 4.7** (Regular atlas). A atlas $\{(U_i, \phi_i)\}_{i \in [E]}$ of compact manifold $\mathcal{M}$ is called regular, if each map $\phi_i$ has the smoothness $\kappa(\phi_i) > \frac{D+3}{2}$.

## 4.2 Theoretical results and insights

**Theorem 4.8** (Main result). *Let $\mathcal{M}$ be a* compact, *$d$-dimensional smooth manifold in $\mathbb{R}^D$, with a regular atlas $\{(U_i, \phi_i)\}_{i \in [E]}$ (Definition 4.7). Let the target function $f : \mathcal{M} \to \mathbb{R}$. Then for any $m \geqslant \Omega(E^2)$, there exists a depth-2 MoE network $\Psi \in \mathcal{H}_{3,m}^{2,E}$, with $E$ experts per layer, each being is a 3-layer $m$-width dense networks, such that:*

$$\|f - \Psi\|_{L_\infty(\mathcal{M})} \leqslant \max_{i \in [E]} \mathcal{E}_{3,m}^{\text{FFN}}(f|_{U_i})$$

$$\leqslant \max_{i \in [E]} \underbrace{\mathcal{E}_{2,m}^{\text{FFN}}\left(f|_{U_i} \circ \phi_i^{-1}\right)}_{\text{approximate a \textcolor{red}{low-dimensional} function}} + \max_{i \in [E]} \underbrace{\mathcal{E}_{2,m}^{\text{FFN}}(\phi_i)}_{\text{approximate a \textcolor{red}{high-order smooth} map}} \left\|f|_{U_i} \circ \phi_i^{-1}\right\|_{\mathcal{C}^1([0,1]^d)}$$

$$\leqslant \max_{i \in [E]} \tilde{\mathcal{O}}\left(m^{-\frac{\kappa(f|_{U_i})}{d} \wedge \frac{1}{2}}\right).$$

Theorem 4.8 demonstrates that depth-2 MoE networks can efficiently approximate functions supported on low-dimensional manifolds. The total approximation error decomposes into two components: (i) approximation of low-dimensional target functions $f|_{U_i} \circ \phi_i^{-1}$'s; (ii) approximation of smooth coordinate maps $\phi_i$'s. Both subproblems are significantly simpler than approximating the original high-dimensional function directly, enabling MoE networks to overcome the curse of dimensionality.

Table 1: Comparison of approximation rates between shallow MoE and shallow dense networks. For MoE, $m$ is the width of each expert networks; for dense networks, $m$ is the width of the hidden layer.

| Shallow MoE networks (Theorem 4.8) | Shallow dense networks (Theorem 3.1) |
|---|---|
| $\max\limits_{i \in [E]} \tilde{\mathcal{O}}\left(m^{-\frac{\kappa(f|_{U_i})}{d} \wedge \frac{1}{2}}\right)$ | $\tilde{\mathcal{O}}\left(m^{-\frac{\kappa(f)}{D} \wedge \frac{1}{2}}\right)$ |

**Improved efficiency over dense networks.** Table 1 highlights the superior approximation efficiency of MoEs. In the regime where the target $f$ has limited smoothness, i.e., $\kappa(f) \ll D$, dense networks suffer from the curse of dimensionality, as the approximation rate $\kappa(f)/D$ deteriorates with ambient dimension $D$. In contrast, MoE networks achieve rates governed by the intrinsic dimension $d \ll D$ and local smoothness $\kappa(f|_{U_i}) \geqslant \kappa(f)$, thereby substantially improving approximation efficiency and achieving faster approximation rate: $\frac{\kappa(f|_{U_i} \circ \phi_i^{-1})}{d} \wedge \frac{1}{2} \gg \frac{\kappa(f)}{D}$.

**Key insight.** The proof of Theorem 4.8 (deferred to Appendix A) reveals several insights into the mechanisms of MoE networks:

*MoE networks achieve efficient approximation by decomposing a complex approximation problem into multiple localized approximation subproblems, as well as a simple assignment task.*

Recall that a depth-2 MoE (Eq. (1)) comprises expert networks and a gating mechanism. Their distinct roles are as follows:

- **Expert networks** ($f^{(2,i)}$ in Layer 2): these components *efficiently approximate the $d$-dimensional local target function $f|_{U_i} \circ \phi_i^{-1}$ and the smooth chart map $\phi_i$*. They directly influence the overall approximation error and benefit from increased width $m$.

- **Routing mechanism** (Layer 1 and gating in Layer 2): these components work together to *exactly assign each input to its correct expert $f^{(2,i)}$*. The first MoE layer $h^{(1)}$ behaves like a dense model, approximating the smooth partition functions $\rho_i$'s. Then the Layer-2 gating network $g^{(2)}$ selects the expert corresponding to the region $U_i$ such that $x \in U_i$. This *exact* assignment is nontrivial, please refer to the proof for details.

**Corollary 4.9** (Special case, highly regular atlas). *Let $\mathcal{M}$ be a compact, $d$-dimensional smooth manifold in $\mathbb{R}^D$, with a highly regular atlas $\{(U_i, \phi_i)\}_{i \in [E]}$ (Example 4.6). Let the target function $f : \mathcal{M} \to \mathbb{R}$. Then for any $m \geqslant \Omega(E^2)$, there exists a depth-2 MoE network $\Psi \in \mathcal{H}_{2,m}^{2,E}$, with $E$ experts per layer, each a 2-layer $m$-width dense network, such that:*

$$\|f - \Psi\|_{L_\infty(\mathcal{M})} \leqslant \max_{i \in [E]} \underbrace{\mathcal{E}_{2,m}^{\mathrm{FFN}}\big(f|_{U_i} \circ \phi_i^{-1}\big)}_{\textit{approximate a \color{red}{low-dimensional} function}} \leqslant \max_{i \in [E]} \tilde{\mathcal{O}}\Big(m^{-\frac{\kappa(f|_{U_i})}{d} \wedge \frac{1}{2}}\Big).$$

Compared to Theorem 4.8, Corollary 4.9 applies to a more structured setting in which the local coordinate maps $\phi_i$ are linear and thus thus do not require approximation. This eliminates the second term in the error bound, and reduces each expert to a 2-layer dense network.

**Comparison with prior works** Shaham et al. (2018); Chen et al. (2019). These works show that *dense networks* can also efficiently approximate functions on low-dimensional manifolds. However, their analyses require additional regularity assumptions on manifolds, which enables explicit constructions of coordinate charts and partition functions. In contrast, our Theorem 4.8 does not require such explicit formulations, applying to a broader class of smooth regular manifolds. More importantly, MoEs offer a fundamental **computational advantage**: while dense networks activate all parameters for every input, MoEs selectively activate only a single expert per input. To achieve a comparable approximation accuracy, the number of activated parameters in dense networks is roughly $E$ times greater than that in MoEs, as dense networks must simultaneously approximate all $E$ subproblems.

## 4.3 Practical Suggestions

Beyond theoretical guarantees, our analysis also offers several practical suggestions into the design of MoE architectures.

**Incorporating nonlinearity into gating is critical.** Our theoretical results indicate that accurate input-to-expert assignment requires approximating the partition functions $\rho_i$, which are generally nonlinear. Since standard gating function is linear and lacks the capacity to model nonlinear $\rho_i$, an additional MoE (or dense) layer is needed prior to gating to approximate $\rho_i$. If the router incorporates sufficient nonlinearity, e.g., a two-layer ReLU routing network, it can directly model complex partitions, reducing the depth and number of parameters. For instance, in Theorem 4.8, the required depth will reduce from 2 to 1, because the nonlinearity in router eliminates the need for a preceding MoE layer. Similarly, in Theorem 5.2, the required depth will reduce from $2L$ to $L$. Therefore, a more direct and potentially more efficient alternative is to **incorporate nonlinearity directly into the gating network**, eliminating the need for a preceding MoE layer. This observation is consistent with recent empirical findings (Zuo et al., 2021; Liu et al., 2022; Nguyen et al., 2024; Akbarian et al., 2024; Le et al., 2024), which demonstrate that nonlinear gating functions improve MoE performance.

**Alternating MoE architectures with equivalent expressive power.** In our construction, the first MoE layer actually serves as a standard dense layer with width $\Omega(E^2)$ to approximate the partition functions $\rho_i$. This insight motivates alternative architectures with comparable expressive power: (i) MoE-dense alternating networks. A natural variant consists of alternating dense and MoE layers, e.g., $h^{\mathrm{moe}} \circ h^{\mathrm{dense}}$. This design extends naturally to deeper architectures. This design has been adopted in practice by GShard (Lepikhin et al., 2020) and GLAM (Du et al., 2022). (ii) Shared + routed experts per MoE layer. Another common MoE variant incorporates one shared expert alongside $E$ routed experts. This structure is empirically adopted in modern MoE architectures such as Qwen2 (Yang et al., 2024b) and DeepSeek (Liu et al., 2024).

**Low-dimensional expert networks via autoencoding.** Our analysis also suggests a more structured and interpretable design for expert networks in MoE. Typically, each expert is implemented as a dense

network with input dimension $D$ and width $\mathcal{O}(D)$, resulting in $\mathcal{O}(D^2)$ parameters. However, our theory motivates replacing each expert $f^{(2l,i)}$ with a composition $f_{\text{low}} \circ \text{Enc}$, where: $\text{Enc} : \mathbb{R}^D \to \mathbb{R}^d$ is an encoder approximating the smooth coordinate chart $\phi : U_i \to [0,1]^d$, $f_{\text{low}} : \mathbb{R}^d \to \mathbb{R}$ is a low-dimensional dense network approximating $f_{l,i} \circ \phi^{-1}$. This design reduces the number of trainable parameters in each expert to $\#(\text{Enc}) + \mathcal{O}(d^2)$, which is significantly smaller than $\mathcal{O}(D^2)$ when $d \ll D$. Moreover, this decomposition aligns with the manifold structure of the target function, improving interpretability. To support encoder learning, one can incorporate a standard reconstruction loss $\mathbb{E}_{\boldsymbol{x}} \|\text{Dec}(\text{Enc}(\boldsymbol{x})) - \boldsymbol{x}\|_2^2$ using a decoder $\text{Dec}$. We leave empirical validation of this theoretically motivated architecture for future work.

# 5 Theory for Multi-layer MoE Networks

**Piecewise functions as multiple tasks.** Modern LLMs are capable of performing a wide range of tasks, such as mathematics, logical reasoning, language understanding, and code generation. From a mathematical perspective, each task can be viewed as a function defined on a task-specific input domain. In practice, these tasks often differ, which are supported on distinct regions $\Omega_1, \cdots, \Omega_N$, with distinct corresponding tasks $f_i : \Omega_i \to \mathbb{R}$. Therefore, performing $N$ tasks can be naturally modeled as approximating a piecewise function: $f(\boldsymbol{x}) = f_i(\boldsymbol{x})$, if $\boldsymbol{x} \in \Omega_i, i \in [N]$. While each $f_i$ may be high-order smooth within its region $\Omega_i$, the global function $f$ may exhibit only low-order smoothness at the interfaces between adjacent regions.

**Key question.** Theorem 4.8 shows that a depth-2 MoE network with $E$ experts per layer can efficiently approximate a piecewise function $f$ comprising $E$ pieces ($f|_{U_i}, i \in [E]$) (each handled by a different expert). A natural question then arises for deep MoE networks:

*How many distinct pieces can be efficiently modeled by a deep MoE network?*

**A naive limitation.** As illustrated above, it is intuitive to associate each expert with a distinct task. A depth-$L$ MoE with $E$ experts per layer contains $\mathcal{O}(LE)$ experts in total, implying a capacity to model at most $\mathcal{O}(LE)$ distinct regions if each expert is used independently.

**Overview of our result: beyond the naive limitation.** Surprisingly, this limitation can be overcome when the target function exhibits *compositional sparsity*. We will show that:

*Depth-$\mathcal{O}(L)$ MoE networks with $E$ experts per layer can efficiently approximate a piecewise function comprising $E^L$ pieces, provided the function satisfies a compositional sparsity structure.*

This demonstrates that MoE networks can model an **exponential number** of structured tasks (far surpassing the native limitation $\mathcal{O}(LE)$) by exploiting structured sparsity in the function.

## 5.1 Piecewise function with compositional sparsity

**Warm-up: Piecewise function on $E^L$ unit cubes with compositional sparsity.** Let the domain be $\mathcal{M} = [0, E]^L$, naturally partitioned into $E^L$ unit cubes. Consider the following target function:

$$f(\boldsymbol{x}) = (f_{1,i_1}(x_1), f_{2,i_2}(x_2), \cdots, f_{L,i_L}(x_L)), \quad (4)$$
$$\text{where } i_l \in \{j \in [E] : x_l \in [j-1, j]\}, \ \forall l \in [L].$$

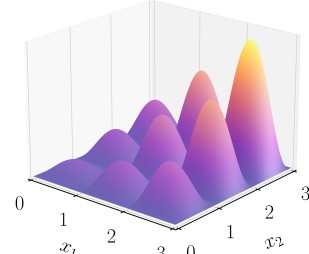

Figure 3: Illustration of Eq. (4): a piece-wise function $f$ with compositional sparsity on $3^2 = 9$ unit cubes. The function $f(\boldsymbol{x}) = f_{1,i_1}(x_1)f_{2,i_2}(x_2)$ is composed from 6 sub-functions: $f_{1,i}(z) = i(i-1-z)(z-i)$ and $f_{2,i}(z) = i(i-1-z)^2(z-i)^2$ on $z \in [i-1, i]$ for $i \in [3]$. Although $f$ is smooth within each region, it is only 0-order continuous on $[0,3] \times [0,3]$.

Here, each subfunction (subtask) $f_{l,i}$ is defined on the interval $U_{l,i} = [i-1, i]$, and $f$ is defined piecewise over $E^L$ regions: $U_{i_1} \times \cdots \times U_{i_L}$, where $i_l \in [E], l \in [L]$).

The function $f$ in Eq. (4) exhibits both:

- **Sparsity**: each subfucntion $f_{l,i}$ depends only on the coordinate $x_l$, not on the full input $\boldsymbol{x}$.

- **Compositionality**: the function $f$ is a hierarchical composition of $L$ selective subfunctions.

Notably, although there are only $LE$ subfunctions, their composition yields $E^L$ distinct functions across the domain.

**Product manifold.** We now extend the above formulation to on general manifolds. Consider a product manifold $\mathcal{M} = \mathcal{M}_1 \times \mathcal{M}_2 \times \cdots \times \mathcal{M}_L$, where each $\mathcal{M}_l$ is a compact $d_l$-dimensional manifold in $\mathbb{R}^D$, and $\sum_{l=1}^{L} d_l \leqslant D$. Each input $\boldsymbol{x} \in \mathcal{M}$ can be written as $\boldsymbol{x} = (\boldsymbol{x}_1, \cdots, \boldsymbol{x}_L)$ with $\boldsymbol{x}_l \in \mathcal{M}_l$.

By compactness, each submanifold $\mathcal{M}_l$ admits a finite smooth atlas $\{(U_{l,i}, \phi_{l,i})\}_{i \in [E_l]}$ and an associated partition of unity $\{\rho_{l_i}\}_{i \in [E_l]}$ (by Theorem 4.4). Without loss of generality, we can let $E_1 = \cdots = E_L$, denoted by $E$.

**General form: piecewise function on product manifold with compositional sparsity.** Consider the target function class admit the form[1]:

$$f(\boldsymbol{x}) = f_{\text{out}}(f_{1,i_1}(\boldsymbol{x}_1), f_{2,i_2}(\boldsymbol{x}_2), \cdots, f_{L,i_L}(\boldsymbol{x}_L)), \quad \text{where}$$
$$i_l \in \{j \in [E] : \boldsymbol{x}_l \in U_{l,j}\}, \ \forall l \in [L]. \tag{5}$$

Here, each subfunction (subtask) $f_{l,i}$ is defined on the local subregion $U_{l,i} \subset \mathcal{M}_l$, and $f_{\text{out}}$ composes their outputs. This extends Eq. (4) from Euclidean coordinates $x_l \in [0, E]$ to manifold-based coordinates $\boldsymbol{x}_l \in \mathcal{M}_l$. Since $f_{\text{out}}$ can typically be approximated by a dense neural network, we assume $f_{\text{out}} = \text{id}$ (the identity map) for simplicity.

We now illustrate this formulation with a concrete example.

Table 2: Semantic interpretation of subregions in Example 5.1

| $\mathcal{M}_1$: Math domain | $\mathcal{M}_2$: Language domain |
|---|---|
| $U_{1,1}$: Geometry | $U_{2,1}$: English |
| $U_{1,2}$: Algebra | $U_{2,2}$: French |
| $U_{1,3}$: Analysis | $U_{2,3}$: German |

**Example 5.1.** Let $\mathcal{M} = \mathcal{M}_1 \times \mathcal{M}_2$, where each $\mathcal{M}_l$ is partitioned into three subregions: $\mathcal{M}_1 = U_{1,1} \cup U_{1,2} \cup U_{1,3}$, $\mathcal{M}_2 = U_{2,1} \cup U_{2,2} \cup U_{2,3}$, with the interpretations given in Table 2. Each subfunction $f_{1,i}$ solves the a specific type of math problem (e.g., geometry), while each $f_{2,i}$ handles text comprehension in a specific language (e.g., English). The full function $f$ defined via Eq. (5) encodes $3 \times 3 = 9$ compositional tasks of the form:

"Understand and solve the [language type] [math type] problem it".

For example, if $\boldsymbol{x}_1 \in U_{1,1}$ and $\boldsymbol{x}_2 \in U_{2,1}$, then $f$ corresponds to the task "understand and solve the English geometry problem".

## 5.2 Theoretical results and insights

We now present our main theoretical result regarding the expressive power of deep MoE networks for approximating piecewise functions with compositional sparsity.

**Theorem 5.2** (Main result). *Let the target function $f$ be of the form* (5)*, which comprises $E^L$ pieces. For each $l \in [L]$, assume that the atlas $\{(U_{l,i}, \phi_{l,i})\}_{i \in [E]}$ of $\mathcal{M}_l$ is regular (Definition 4.7). Then there exists a depth-$2L$ MoE network $\Psi \in \mathcal{H}_{3,m}^{2L,E}$ with $m \geqslant \Omega(E^2)$, such that:*

$$\|f - \Psi\|_{L_\infty(\mathcal{M})} \leqslant \max_{l \in [L]} \max_{i \in [E]} \mathcal{E}_{3,m}^{\text{FFN}}(f_{l,i})$$

$$\leqslant \max_{l \in [L]} \max_{i \in [E]} \underbrace{\mathcal{E}_{2,m}^{\text{FFN}}(f_{l,i} \circ \phi_{l,i}^{-1})}_{\text{approximate a {\color{red}low-dimensional} function}} + \max_{l \in [L]} \max_{i \in [E_l]} \underbrace{\mathcal{E}_{2,m}^{\text{FFN}}(\phi_{l,i})}_{\text{approximate a {\color{red}smooth} map}} \left\| f_{l,i} \circ \phi_{l,i}^{-1} \right\|_{\mathcal{C}^1([0,1]^{d_l})}$$

$$\leqslant \max_{l \in [L]} \max_{i \in [E]} \tilde{\mathcal{O}}\left(m^{-\frac{\kappa(f_{l,i})}{d_l} \wedge \frac{1}{2}}\right).$$

---
[1]To ensure $f$ is well-defined on overlapping charts, we assume $f_{l,i}(\boldsymbol{x}_l) = f_{l,j}(\boldsymbol{x}_l), \forall \boldsymbol{x}_l \in U_i \cap U_j, i \neq j$.

Theorem 5.2 establishes that a depth-$2L$ MoE network with $E$ experts per layer can efficiently approximate a piecewise function with $E^L$ pieces, provided the function exhibits compositional sparsity. The approximation error consists of two components: (i) approximation of local low-dimensional subfunctions $f_{l,i} \circ \phi_{l,i}^{-1}$; (ii) approximation of smooth coordinate maps $\phi_{l,i}$. The resulting approximation rate is $\max_{l \in [L]} \max_{i \in [E]} \tilde{\mathcal{O}}\left(m^{-\frac{\kappa(f_{l,i})}{d_l} \wedge \frac{1}{2}}\right)$, which avoids the curse of dimenisonality. In the special case $L = 1$, this result recovers Theorem 5.2.

**Key insight.** The proof (deferred to Appendix B) reveals the following intuitions:

*Each pair of MoE layers implements $E$ subtasks,*
*and a depth-$2L$ architecture enables hierarchical composition of $E^L$ tasks.*

- For each $l \in [L]$, the $(2l-1)$-st and $2l$-th layers approximate the $E$ subfunctions $f_{l,i}, (i \in [E])$ defined on the manifold charts $U_{l,i} \in \mathcal{M}_i$ and assign $\boldsymbol{x}_l$ to their correct experts, following the same mechanism as in Theorem 4.8.
- Stacking $L$ such MoE blocks composes these representations hierarchically, approximating the final function $(f_{1,i_1}(\boldsymbol{x}_1), \cdots, f_{L,i_L}(\boldsymbol{x}_L))$.

**Illustration via Example 5.1.** In Example 5.1 with $L = 2$ and $E = 3$ (3 subregions for each of math and language). Theorem 5.2 shows that a depth-4 MoE network with 3 experts per layer can express all $3 \times 3 = 9$ tasks of the form "understand and solve the [language type] [math type] problem". Concretely,

- Experts in Layer 2 (math) implement $f_{2,i}$ for $i = 1, 2, 3$, corresponding to "solving `geometry`/`algebra`/`analysis` problems", respectively..
- Experts in Layer 4 (language) implement $f_{4,i}$ for $i = 1, 2, 3$, corresponding to "understanding `English`/`French`/`German`", respectively.
- Layer 1 and the gating in Layer 2 together perform routing for $\boldsymbol{x}_1$; Layer 3 and the gating in Layer 4 together perform routing for $\boldsymbol{x}_2$.

**Theorem 5.3** (Warmup case). *Let the target function $f$ be of the form* (4). *Then there exists a depth-$2L$ MoE network $\Psi \in \mathcal{H}_{2,m}^{2L,E}$ with $m \geqslant \Omega(E)$, such that:*

$$\|f - \Psi\|_{L_\infty(\mathcal{M})} \leqslant \max_{l \in [L]} \max_{i \in [E]} \underbrace{\mathcal{E}_{2,m}^{\text{FFN}}\left(f_{l,i}\right)}_{\textit{approximate a 1-dimensional function}} \leqslant \max_{l \in [L]} \max_{i \in [E]} \mathcal{O}\left(m^{-\kappa(f_{l,i}) \wedge \frac{1}{2}}\right).$$

Compared to Theorem 5.2, this result requires only: (i) shallower expert networks (2-layer instead of 3-layer), (ii) smaller expert width ($m \geqslant \Omega(E)$ instead of $\Omega(E^2)$) due to the simple geometry (Euclidean cubes).

Although each subfunction may have high-order smoothness ($\kappa(f_{l,i}) \gg 1$), the composite function $f$ can exhibit only low-order smoothness (e.g., $\kappa(f) = 0, 1$) due to low-order regularity at the interfaces between adjacent regions. As a result, directly approximating the global function $f$ is inefficient. A natural and efficient approach is to decompose the approximation problem into localized subproblems, each defined on a simple subregion with high regularity. Our constructive MoE networks can provably achieve this approach.

# 6 Experimental Validation

To support our main theoretical results, we conduct two new experiments, each aligned with one of our key insights. The experimental details are shown in Appendix C.

**Experiment I. Shallow MoEs for low-dimensional functions.** To validate our theoretical insight in Section 4 (Theorem 4.8): shallow MoE networks can efficiently approximate functions supported on low-dimensional manifolds and *overcome the curse of dimensionality*.

Specifically, we consider the low-dimensional manifold $\mathcal{M} = \{\boldsymbol{x} \in \mathbb{R}^D : x_1^2 + x_2^2 = 1; x_i = 0, \forall i > 2\}$ embedded in $\mathbb{R}^D$ with $D > 2$. The target function is $f(\boldsymbol{x}) = \sin(5x_1) + \cos(3x_2)$, defined on $\mathcal{M}$. As a model, we consider "1-4-MoE", a 1-layer MoE comprising 1 router and 4 experts, where each

expert is a two-layer ReLU network with hidden width $10$. To validate whether MoE can overcome the curse of dimensionality, we vary the input dimension $D \in \{16, 32, 64, 128\}$.

As shown in Table 3, one can see that: as $D$ increases, the test error of MoE does not increase significantly and remains stable. This supports our insight that shallow MoEs efficiently approximate functions on low-dimensional manifolds and avoid the curse of dimensionality.

Table 3: (Results of Experiment I) The test error of 1-4-MoE under different input dimensions $D$.

| input dim $D$ | 16 | 32 | 64 | 128 |
|---|---|---|---|---|
| test error | 3.40e-4 | 3.38e-4 | 3.17e-4 | 3.42e-4 |

**Experiment II. Deep MoEs for piecewise functions.** To verify our theoretical insight in Section 5: depth-$\mathcal{O}(L)$ MoE networks with $E$ experts per layers can efficiently approximate piecewise functions with $E^L$ distinct pieces.

As defined in our Figure 3, we consider the piecewise function $f$ with compositional sparsity defined over $3^2 = 9$ unit cubes. As the model, we consider "2-3-MoE" (a 2-layer MoE comprising 2 routing layers and 2 expert layers with 3 experts each); To illustrate the role of depth, we also consider a shallow "1-6-MoE", with comparable parameter count. Each expert is a two-layer ReLU FFN with hidden width $m \in \{16, 32, 64, 128\}$. To validate whether 2-3-MoE and 1-6-MoE can approximate this target, we vary the hidden width $m$.

The results, shown in Table 4, illustrate that: *(i)* As $m$ increases, 2-3-MoE achieves rapidly decreasing error. This supports that the depth-2 MoE with 3 experts per layers can efficiently approximate this piecewise function with $3^2$ distinct pieces; *(ii)* In contrast, 1-6-MoE exhibits a performance plateau, revealing its limited expressive power. This highlights the crucial role of depth in modeling such compositional structures.

Table 4: (Results of Experiment II) The test error of the 2-3-MoE and 1-6-MoE under different hidden width $m$.

| hidden width $m$ of experts | 16 | 32 | 64 | 128 |
|---|---|---|---|---|
| test error of 2-3-MoE | 8.32e-5 | 1.41e-5 | 4.73e-6 | 2.59e-6 |
| test error of 1-6-MoE | 7.96e-5 | 2.17e-5 | 2.65e-5 | 4.60e-5 |

# 7 Conclusion and Future Work

In this work, we provide a theoretical study of the expressive power of MoE networks for modeling structured complex tasks. For shallow MoE networks, we show that they can efficiently approximate functions supported on low-dimensional manifolds, overcoming the curse of dimensionality. For deep MoE networks, we establish that, when the target exhibits a compositional sparsity structure, they can approximate piecewise functions consisting of exponentially many distinct pieces, effectively modeling an exponential number of tasks.

**Beyond compositional sparsity.** Our analysis focuses on compositional sparsity, a structure that is both theoretically rich and practically relevant. However, real-world problems may involve other forms of sparsity, such as group sparsity, graph sparsity, or temporal sparsity. Characterizing the expressive power of deep MoE networks under these alternative structures is an important direction for future work.

**Training dynamics analysis.** While our study addresses the approximation capabilities of MoE networks, a critical open question is whether such expressive solutions can be found via training algorithms such as stochastic gradient descent. Analyzing the training dynamics of MoE networks is considerably more challenging than in dense architectures, due to auxiliary load-balancing objectives and the use of non-differentiable Top-$K$ routing.

# Acknowledgments

Mingze Wang is supported by Young Scientists (PhD) Fund of the National Natural Science Foundation of China (No. 124B2028).

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

# Appendix

# A  Proofs in Section 4

## A.1  Further Introduction to Manifolds

Compact smooth manifolds $\mathcal{M} \subset \mathbb{R}^D$ are known to have strong regularity properties.

We begin by introducing the definition of the reach, a geometric quantity affected by two factors: the curvature of the manifold and the width of the narrowest bottleneck-like structure of $\mathcal{M}$.

**Definition A.1** (reach). Let $\mathcal{A}(\mathcal{M}) := \{z \in \mathbb{R}^D : \exists p \neq q \in \mathcal{M}, \|p - z\|_2 = \|q - z\|_2 = \inf_{y \in \mathcal{M}} \|y - z\|_2\}$ be the set of points that have at least two nearest neighbors on $\mathcal{M}$. Then the reach of $\mathcal{M}$ is defined as $\tau_{\mathcal{M}} := \inf_{z \in \mathcal{M}, y \in \mathcal{A}(\mathcal{M})} \|y - z\|_2$.

The following lemma has established that a compact smooth manifold has positive reach.

**Lemma A.2** (Federer (1959)). *Let $\mathcal{M}$ be a compact smooth manifold in $\mathbb{R}^D$. Then it has positive reach, i.e., $\tau_{\mathcal{M}} > 0$.*

If $\mathcal{M}$ has the reach greater than $\tau > 0$, then intuitively, one can roll freely a ball of radius $\tau$ around it. This ensures the existence of a well-structured atlas of compact smooth manifold in Example 4.6:

**Highly regular atlas of compact smooth manifold $\mathcal{M}$.** (Chen et al., 2019) Let $r > 0$ and consider the open cover $\{\mathbb{B}(x; r)\}_{x \in \mathcal{M}}$ of $\mathcal{M}$, where $\mathbb{B}(x; r)$ denotes the $\ell_2$ Euclidean ball of radius $r$ centered at $x$. Since $\mathcal{M}$ is compact, there exists a finite subcover such that $\mathcal{M} \subset \cup_{i \in [E]} \mathbb{B}(c_i; r)$, where $c_i \in \mathcal{M}$. Now we pick the radius $r < \tau_{\mathcal{M}}/2$ such that

$$U_i = \mathcal{M} \cap \mathbb{B}(c_i; r)$$

is diffeomorphic to a ball in $\mathbb{R}^d$ (Niyogi et al., 2008). We denote the tangent space at $c_i$ as $\mathcal{T}_{c_i}(\mathcal{M}) = \text{span}(v_{i,1}, \cdots, v_{i,d})$, where $\{v_{i,1}, \cdots, v_{i,d}\}$ are orthonormal basis. Define the matrix $V_i = [v_{i,1}, \cdots, v_{i,d}] \in \mathbb{R}^{D \times d}$ and construct the chart map as:

$$\phi_i(x) = b_i(V_i^\top(x - c_i) + s_i) \in [0, 1]^d, \quad \forall x \in U_i,$$

where $b_i \in (0, 1]$ is a scaling factor and $s_i$ is a translation vector chosen such that $\phi_i(x) \in [0, 1]^d$. Then $\{(U_i, \phi_i)\}_{i \in [E]}$ is a smooth atlas on $\mathcal{M}$. We call it highly regular, since (i) each $U_i$ is diffeomorphic to a ball in $\mathbb{R}^d$; (ii) each chart map $\phi_i$ is a linear function, implying infinite smoothness.

## A.2 Proof of Theorem 4.8

For simplicity, we use the notation $\|\cdot\|$ to denote the $L_\infty$ norm in the proof.

**Routing mechanism (Layer 1 and gating in Layer 2).**

*Proof sketch:* These components work together to *exactly assign each input $x$ to its correct expert* $f^{(2,i)}$. Specifically, the first MoE layer $h^{(1)}$ behaves like a dense model, approximating the smooth partition functions $\rho_i$'s. Then the Layer-2 gating network $g^{(2)}$ selects the expert corresponding to the region $U_i$ such that $x \in U_i$.

By Theorem 3.1, there exists $E$ two-layer networks $\tau_i$ $(i \in [E])$, with width $m_i \geqslant \Omega(E^2)$, such that:

$$\|\rho_i - \tau_i\| \leqslant \frac{1}{4E}, \quad \forall i \in [E].$$

Let $\tau$ be the concatenated two-layer network of the $E$ two-layer networks, i.e., $\tau(x) = (\tau_1(x), \cdots, \tau_E(x))^\top$, then it satisfies:

$$\left\|(\rho_1, \cdots, \rho_E)^\top - \tau\right\| \leqslant \frac{1}{4E}. \tag{6}$$

In Layer 1, each expert network $f^{(1,i)} : \mathbb{R}^D \to \mathbb{R}^{D+E}$ shares the same form:

$$f^{(1,i)}(x) = \begin{pmatrix} x \\ \tau(x) \end{pmatrix}.$$

The weight matrix $W_R^{(1)}$ of the gating network $g^{(1)} : \mathbb{R}^D \to \mathbb{R}^E$ can be any constant, such as $\mathbf{0}$. In fact, Layer 1 behaves as a standard dense network computing $\tau$.

For the gating network $g^{(2)}$ in Layer 2, we define its weight matrix as:

$$W_R^{(2)} = \left(\mathbf{0}_{E \times D}, I_{E \times E}\right) \in \mathbb{R}^{E \times (D+E)}.$$

Now we check the routing mechanism: given an input $x \in \mathcal{M}$, Layer 1 outputs

$$h^{(1)}(x) = \begin{pmatrix} x \\ \tau(x) \end{pmatrix} \in \mathbb{R}^{D+E}.$$

Then the gating output of the Layer 2 is

$$g^{(2)}(h^{(1)}(x)) = W_R^{(2)} h^{(1)}(x) = \tau(x) \in \mathbb{R}^E.$$

Then it will assign $x$ to the $i_x$-th expert network in Layer 2, where

$$i_x := \arg\max_{i \in [E]} g_i^{(2)}(x) = \arg\max_{i \in [E]} \tau_i(x).$$

Since $\sum_{i \in [E]} \rho_i(x) = 1$ and $\rho_i \geqslant 0, \forall i \in [E]$, we must have $\max_{i \in [E]} \rho_i(x) \geqslant \frac{1}{E}$. Using (6), we further have:

$$\tau_{i_x}(x) = \max_{i \in [E]} \tau_i(x) \geqslant \max_{i \in [E]} \rho_i(x) - \frac{1}{4E} = \frac{3}{4E},$$

which implies

$$\rho_{i_x}(x) \geqslant \tau_{i_x}(x) - \frac{1}{4E} \geqslant \frac{3}{4E} - \frac{1}{4E} = \frac{1}{2E} > 0.$$

Thus,

$$x \in U_{i_x}. \tag{7}$$

This establishes exact assignment of $x$ to its correct region.

**Expert networks (Layer 2).**

*Proof sketch:* Each expert network $f^{(2,i)}$ in Layer 2 is responsible for *approximating the d-dimensional local target functions* $f|_{U_i} \circ \phi_i^{-1}$ *and the smooth chart maps* $\phi_i$.

By Theorem 3.1, there exist two-layer dense networks $g_i$ and $\psi_i$, with width $m$, such that

$$\left\| f|_{U_i} \circ \phi_i^{-1} - g_i \right\| \leqslant \tilde{\mathcal{O}} \left( \max \left\{ m^{-\frac{\kappa(f|_{U_i} \circ \phi_i^{-1})}{d}}, m^{-1/2} \right\} \right) \leqslant \tilde{\mathcal{O}} \left( m^{-\frac{\kappa(f|_{U_i} \circ \phi_i^{-1})}{d} \wedge \frac{1}{2}} \right);$$

$$\left\| \phi_i - \psi_i \right\| \leqslant \tilde{\mathcal{O}} \left( m^{-1/2} \right).$$

Let the expert network $f^{(2,i)} : \mathbb{R}^{D+E} \to \mathbb{R}$ as

$$f^{(2,i)}(h^{(1)}(\boldsymbol{x})) := g_i \circ \psi_i(h^{(1)}_{1:E}(\boldsymbol{x})) = g_i \circ \psi_i(\boldsymbol{x}),$$

which is a three-layer dense network and satisfies:

$$\begin{aligned}
\sup_{\boldsymbol{x} \in U_i} \left| f|_{U_i}(\boldsymbol{x}) - f^{(2,i)}(h^{(1)}(\boldsymbol{x})) \right| &= \sup_{\boldsymbol{x}} \left| f|_{U_i} \circ \phi_i^{-1} \circ \phi_i(\boldsymbol{x}) - g_i \circ \psi_i(\boldsymbol{x}) \right| \\
&= \left\| f|_{U_i} \circ \phi_i^{-1} \circ \phi_i - f|_{U_i} \circ \phi_i^{-1} \circ \psi_i + f|_{U_i} \circ \phi_i^{-1} \circ \psi_i - g_i \circ \psi_i \right\| \\
&\leqslant \left\| f|_{U_i} \circ \phi_i^{-1} \circ \phi_i - f|_{U_i} \circ \phi_i^{-1} \circ \psi_i \right\| + \left\| f|_{U_i} \circ \phi_i^{-1} \circ \psi_i - g_i \circ \psi_i \right\| \\
&\leqslant \left\| f|_{U_i} \circ \phi_i^{-1} \right\| \left\| \phi_i - \psi_i \right\| + \left\| f|_{U_i} \circ \phi_i^{-1} - g_i \right\| \\
&\leqslant \tilde{\mathcal{O}} \left( m^{-\frac{\kappa(f|_{U_i} \circ \phi_i^{-1})}{d} \wedge \frac{1}{2}} \right) = \tilde{\mathcal{O}} \left( m^{-\frac{\kappa(f|_{U_i})}{d} \wedge \frac{1}{2}} \right).
\end{aligned} \tag{8}$$

### The final estimate.

Given an input $\boldsymbol{x} \in \mathcal{M}$, the routing mechanism assigns it to the correct expert network $f^{(2,i_{\boldsymbol{x}})}$, which satisfies $\boldsymbol{x} \in U_{i_{\boldsymbol{x}}}$ (7). Consequently, the approximation error holds:

$$|f(\boldsymbol{x}) - f^{(2,i_{\boldsymbol{x}})}(h^{(1)}(\boldsymbol{x}))| = \left| f|_{U_{i_{\boldsymbol{x}}}}(\boldsymbol{x}) - f^{(2,i_{\boldsymbol{x}})}(h^{(1)}(\boldsymbol{x})) \right|$$

$$\overset{(8)}{\leqslant} \tilde{\mathcal{O}} \left( m^{-\frac{\kappa(f|_{U_{i_{\boldsymbol{x}}}})}{d} \wedge \frac{1}{2}} \right) \leqslant \max_{i \in [E]} \tilde{\mathcal{O}} \left( m^{-\frac{\kappa(f|_{U_i})}{d} \wedge \frac{1}{2}} \right).$$

Since $\boldsymbol{x}$ is arbitrary, this concludes the proof.

Additionally, although this analysis is based on ReLU-activated experts, the theoretical framework extends naturally to MoEs with other activation functions (e.g., GeLU, SiLU, Swish) by using the results in Zhang et al. (2024).

## A.3  Proof of Corollary 4.9

The proof follows exactly the same structure as Theorem 4.8, except for a simplification in the step concerning the **Expert networks (Layer 2)**. We illustrate only the modified part below.

### Expert networks (Layer 2).

*Compared to Section A.2, the key simplification is that the chart maps $\phi_i$ are* linear *(see Example 4.6). Thus, we do not need to approximate them with neural networks. We only need to approximate the composition $f|_{U_i} \circ \phi_i^{-1}$ using a two-layer network.*

By Theorem 3.1, there exists a two-layer dense networks $g_i$, with width $m$, such that

$$\left\| f|_{U_i} \circ \phi_i^{-1} - g_i \right\| \leqslant \tilde{\mathcal{O}} \left( \max \left\{ m^{-\frac{\kappa(f|_{U_i} \circ \phi_i^{-1})}{d}}, m^{-1/2} \right\} \right) \leqslant \tilde{\mathcal{O}} \left( m^{-\frac{\kappa(f|_{U_i} \circ \phi_i^{-1})}{d} \wedge \frac{1}{2}} \right).$$

Let the expert network $f^{(2,i)} : \mathbb{R}^{D+E} \to \mathbb{R}$ as

$$f^{(2,i)}(h^{(1)}(\boldsymbol{x})) := g_i \circ \phi_i(h^{(1)}_{1:E}(\boldsymbol{x})) = g_i \circ \phi_i(\boldsymbol{x}),$$

which is a two-layer dense network since $\phi_i$ is linear and $g_i$ is two-layer. It satisfies:

$$\sup_{\boldsymbol{x} \in U_i} \left| f|_{U_i}(\boldsymbol{x}) - f^{(2,i)}(h^{(1)}(\boldsymbol{x})) \right| = \sup_{\boldsymbol{x}} \left| f|_{U_i} \circ \phi_i^{-1} \circ \phi_i(\boldsymbol{x}) - g_i \circ \phi_i(\boldsymbol{x}) \right|$$

$$= \left\| f|_{U_i} \circ \phi_i^{-1} \circ \phi_i - g_i \circ \phi_i \right\| \leqslant \left\| f|_{U_i} \circ \phi_i^{-1} - g_i \right\| \tag{9}$$

$$\leqslant \tilde{\mathcal{O}} \left( m^{-\frac{\kappa(f|_{U_i} \circ \phi_i^{-1})}{d} \wedge \frac{1}{2}} \right) = \tilde{\mathcal{O}} \left( m^{-\frac{\kappa(f|_{U_i})}{d} \wedge \frac{1}{2}} \right).$$

# B    Proofs in Section 5

## B.1    Proof of Theorem 5.2

The key idea is that for each $l \in [L]$, Layers $2l - 1$ and $2l$ jointly approximate the subfunction

$$f_{l,i_l}(\boldsymbol{x}_l), \quad \text{where } i_l \in \{j \in [E] : \boldsymbol{x}_l \in U_{l,j}\}.$$

The overall network composition then constructs the full function $f$ from these subcomponents, as defined in Equation (5).

**Embedding.** To facilitate layerwise operations, we first embed the input $\boldsymbol{x} = (\boldsymbol{x}_1^\top, \cdots, \boldsymbol{x}_L^\top)^\top \in \mathbb{R}^{LD}$ into an extended space:

$$\boldsymbol{x}^{(0)} = (\boldsymbol{x}_1^\top, \cdots, \boldsymbol{x}_L^\top, \boldsymbol{0}_L^\top)^\top \in \mathbb{R}^{LD+L},$$

where the final $L$ entries are used to store the subfunctions approximated in subsequent layers. And we denote

$$\tilde{D} := LD + L.$$

For simplicity, we denote the output of $2l$-th layer ($l \in [L]$) be $\boldsymbol{x}^{(2l)} \in \mathbb{R}^{\tilde{D}}$.

Each two-layer MoE block operates similarly to the shallow case studied in Section A.2, and thus the core proof strategy carries over. For completeness and clarity, we present the full proof below.

### $l$-th routing mechanism (Layer $2l - 1$ and gating in Layer $2l$).

*Proof sketch:* These components work together to *exactly assign $\boldsymbol{x}_l$ to its correct expert $f^{(2l,i)}$.* Specifically, the first MoE layer $h^{(2l-1)}$ behaves like a dense model, approximating the smooth partition functions $\rho_{l,i}$'s. Then the Layer-2 gating network $g^{(2l)}$ selects the expert corresponding to the region $U_i$ such that $\boldsymbol{x}_l \in U_{l,i}$.

By Theorem 3.1, there exists $E$ two-layer networks $\tau_{l,i}$ ($i \in [E]$), with width $m_{l,i} \geqslant \Omega(E^2)$, such that:

$$\|\rho_{l,i} - \tau_{l,i}\| \leqslant \frac{1}{4E}, \quad \forall i \in [E].$$

Let $\tau^{(l)}$ be the concatenated two-layer network of the $E$ two-layer networks, i.e., $\tau^{(l)}(\boldsymbol{x}_l) = (\tau_{l,i}(\boldsymbol{x}_l), \cdots, \tau_{l,E}(\boldsymbol{x}_l))^\top$, then it satisfies:

$$\left\| (\rho_{l,1}, \cdots, \rho_{l,E})^\top - \tau^{(l)} \right\| \leqslant \frac{1}{4E}. \tag{10}$$

In Layer $2l - 1$, each expert network $f^{(2l-1,i)} : \mathbb{R}^{\tilde{D}} \to \mathbb{R}^{\tilde{D}+E}$ shares the same form:

$$f^{(2l-1,i)}(\boldsymbol{x}^{(2l-2)}) = \begin{pmatrix} \boldsymbol{x}^{(2l-2)} \\ \tau^{(l)}(\boldsymbol{x}_l) \end{pmatrix}.$$

The weight matrix $\boldsymbol{W}_R^{(2l-1)}$ of the gating network $g^{(2l-1)} : \mathbb{R}^{\tilde{D}} \to \mathbb{R}^E$ can be any constant, such as $\boldsymbol{0}$. In fact, Layer $2l - 1$ behaves as a standard dense network computing $\tau^{(l)}(\boldsymbol{x}_l)$.

For the gating network $g^{(2l)}$ in Layer $2l$, we define its weight matrix as:
$$\boldsymbol{W}_R^{(2l)} = \left(\boldsymbol{0}_{E\times\tilde{D}}, \boldsymbol{I}_{E\times E}\right) \in \mathbb{R}^{E\times(\tilde{D}+E)}.$$

Now we check the routing mechanism: given an input $\boldsymbol{x} = (\boldsymbol{x}_1, \cdots, \boldsymbol{x}_L) \in \mathcal{M}$, where $\boldsymbol{x}_l \in \mathcal{M}_l$, Layer $2l-1$ outputs
$$\boldsymbol{x}^{(2l-1)} = h^{(2l-1)}(\boldsymbol{x}^{(2l-2)}) = \begin{pmatrix} \boldsymbol{x}^{(2l-2)} \\ \tau^{(l)}(\boldsymbol{x}_l) \end{pmatrix} \in \mathbb{R}^{\tilde{D}+E}.$$

Then the gating output of the Layer $2l$ is
$$g^{(2l)}(\boldsymbol{x}^{(2l-1)}) = \boldsymbol{W}_R^{(2l)} h^{(2l-1)}(\boldsymbol{x}_l) = \tau^{(l)}(\boldsymbol{x}_l) \in \mathbb{R}^E.$$

Then it will assign $\boldsymbol{x}_l$ to the $i_{\boldsymbol{x}_l}$-th expert network in Layer 2, where
$$i_{\boldsymbol{x}_l} := \underset{i\in[E]}{\arg\max}\, g_i^{(2l)}(\boldsymbol{x}_l) = \underset{i\in[E]}{\arg\max}\, \tau_{l,i}(\boldsymbol{x}_l).$$

Since $\sum_{i\in[E]} \rho_{l,i}(\boldsymbol{x}_l) = 1$ and $\rho_{l,i} \geqslant 0, \forall i \in [E]$, we must have $\max_{i\in[E]} \rho_{l,i}(\boldsymbol{x}_l) \geqslant \frac{1}{E}$. Using (10), we further have:
$$\tau_{i_{\boldsymbol{x}_l}}(\boldsymbol{x}_l) = \max_{i\in[E]} \tau_{l,i}(\boldsymbol{x}_l) \geqslant \max_{i\in[E]} \rho_{l,i}(\boldsymbol{x}_l) - \frac{1}{4E} = \frac{3}{4E},$$

which implies
$$\rho_{i_{\boldsymbol{x}_l}}(\boldsymbol{x}_l) \geqslant \tau_{i_{\boldsymbol{x}_l}}(\boldsymbol{x}_l) - \frac{1}{4E} \geqslant \frac{3}{4E} - \frac{1}{4E} = \frac{1}{2E} > 0.$$

Thus,
$$\boldsymbol{x}_l \in U_{i_{\boldsymbol{x}_l}}. \tag{11}$$

This establishes exact assignment of $\boldsymbol{x}_l$ to its correct region.

### $l$-th expert networks (Layer $2l$).

*Proof sketch:* Each expert network $f^{(2l,i)}$ in Layer 2 is responsible for *approximating the d-dimensional local target functions $f_{l,i} \circ \phi_{l,i}^{-1}$ and the smooth chart maps $\phi_{l,i}$.*

By Theorem 3.1, there exist two-layer dense networks $g_{l,i}$ and $\psi_{l,i}$, with width $m$, such that
$$\left\| f_{l,i} \circ \phi_{l,i}^{-1} - g_{l,i} \right\| \leqslant \tilde{\mathcal{O}}\left( \max\left\{ m^{-\frac{\kappa(f_{l,i}\circ\phi_{l,i}^{-1})}{d_l}}, m^{-1/2} \right\} \right) \leqslant \tilde{\mathcal{O}}\left( m^{-\frac{\kappa(f_{l,i}\circ\phi_{l,i}^{-1})}{d_l}\wedge\frac{1}{2}} \right);$$
$$\|\phi_{l,i} - \psi_{l,i}\| \leqslant \tilde{\mathcal{O}}\left( m^{-1/2} \right).$$

Let the expert network $f^{(2l,i)} : \mathbb{R}^{\tilde{D}+E} \to \mathbb{R}^{\tilde{D}}$ as
$$f^{(2l,i)}(\boldsymbol{x}^{(2l-1)}) := \begin{pmatrix} \boldsymbol{x} \\ \boldsymbol{0}_{l-1} \\ g_{l,i}\circ\psi_{l,i}(\boldsymbol{x}_{1+(l-1)E:\ lE}^{(2l-1)}) \\ \boldsymbol{0}_{E-l} \end{pmatrix} = \begin{pmatrix} \boldsymbol{x} \\ \boldsymbol{0}_{l-1} \\ g_{l,i}\circ\psi_{l,i}(\boldsymbol{x}_l) \\ \boldsymbol{0}_{E-l} \end{pmatrix},$$

which is a three-layer dense network and satisfies:
$$\begin{aligned}
\sup_{\boldsymbol{x}_l\in U_{l,i}} \left| f_{l,i}(\boldsymbol{x}_l) - f_{\tilde{D}+l}^{(2l,i)}(\boldsymbol{x}^{(2l-1)}) \right| &= \sup_{\boldsymbol{x}_l\in U_{l,i}} \left| f_{l,i}\circ\phi_{l,i}^{-1}\circ\phi_{l,i}(\boldsymbol{x}_l) - g_{l,i}\circ\psi_{l,i}(\boldsymbol{x}_l) \right| \\
&= \left\| f_{l,i}\circ\phi_{l,i}^{-1}\circ\phi_{l,i} - f_{l,i}\circ\phi_{l,i}^{-1}\circ\psi_{l,i} + f_{l,i}\circ\phi_{l,i}^{-1}\circ\psi_{l,i} - g_{l,i}\circ\psi_{l,i} \right\| \\
&\leqslant \left\| f_{l,i}\circ\phi_{l,i}^{-1}\circ\phi_{l,i} - f_{l,i}\circ\phi_{l,i}^{-1}\circ\psi_{l,i} \right\| + \left\| f_{l,i}\circ\phi_{l,i}^{-1}\circ\psi_{l,i} - g_{l,i}\circ\psi_{l,i} \right\| \\
&\leqslant \left\| f_{l,i}\circ\phi_{l,i}^{-1} \right\| \|\phi_{l,i} - \psi_{l,i}\| + \left\| f_{l,i}\circ\phi_{l,i}^{-1} - g_{l,i} \right\| \\
&\leqslant \tilde{\mathcal{O}}\left( m^{-\frac{\kappa(f_{l,i}\circ\phi_{l,i}^{-1})}{d_l}\wedge\frac{1}{2}} \right) = \tilde{\mathcal{O}}\left( m^{-\frac{\kappa(f_{l,i})}{d_l}\wedge\frac{1}{2}} \right).
\end{aligned} \tag{12}$$

**The final estimate.**

From the above construction, the output of the $2L$-th layer takes the form:

$$\boldsymbol{x}^{(2L)} = \begin{pmatrix} \boldsymbol{x} \\ g_{1,i} \circ \psi_{1,i_{\boldsymbol{x}_1}}(\boldsymbol{x}_1) \\ \vdots \\ g_{L,i} \circ \psi_{L,i_{\boldsymbol{x}_L}}(\boldsymbol{x}_L) \end{pmatrix} \in \mathbb{R}^{LD+L}.$$

We decode this vector by applying a linear projection that extracts the last $L$ components, yielding:

$$\Psi(\boldsymbol{x}) := (\boldsymbol{0}_{L\times LD}, \boldsymbol{I}_{L\times L})\, \boldsymbol{x}^{(2L)} = \begin{pmatrix} g_{1,i} \circ \psi_{1,i_{\boldsymbol{x}_1}}(\boldsymbol{x}_1) \\ \vdots \\ g_{L,i} \circ \psi_{L,i_{\boldsymbol{x}_L}}(\boldsymbol{x}_L) \end{pmatrix} \in \mathbb{R}^L.$$

Now consider an input $\boldsymbol{x} = (\boldsymbol{x}_1, \cdots, \boldsymbol{x}_L) \in \mathcal{M}$. For each $l \in [L]$, the $l$-th routing mechanism assigns it to the correct expert network $f^{(2l,i_{\boldsymbol{x}_l})}$, which satisfies $\boldsymbol{x}_l \in U_{i_{\boldsymbol{x}_l}}$ (11). Consequently, the approximation error holds:

$$\|f(\boldsymbol{x}) - \Psi(\boldsymbol{x})\| = \max_{l \in [L]} \left| f_{l,i_{\boldsymbol{x}_l}}(\boldsymbol{x}_l) - g_{l,i_{\boldsymbol{x}_l}} \circ \psi_{l,i_{\boldsymbol{x}_l}}(\boldsymbol{x}_l) \right|$$

$$\overset{(12)}{\leqslant} \max_{l \in [L]} \tilde{\mathcal{O}}\left( m^{-\frac{\kappa(f_{l,i_{\boldsymbol{x}_l}})}{d_l} \wedge \frac{1}{2}} \right) \leqslant \max_{l \in [L]} \max_{i \in [E]} \tilde{\mathcal{O}}\left( m^{-\frac{\kappa(f_{l,i})}{d_l} \wedge \frac{1}{2}} \right).$$

Since $\boldsymbol{x}$ is arbitrary, this concludes the proof.

## B.2   Proof of Theorem 5.3

The proof follows the same structure as Theorem 5.2, except for a simplification in the construction of the **routing mechanism**. For completeness and clarity, we provide the full proof below.

The key idea is that for each $l \in [L]$, Layers $2l - 1$ and $2l$ jointly approximate the subfunction

$$f_{l,i_l}(x_l), \text{ where } i_l \in \{j \in [E] : x_l \in U_{l,j} = [j-1, j]\}.$$

The overall network composition then constructs the full function $f$ from these subcomponents.

For simplicity, we denote the output of $2l$-th layer ($l \in [L]$) be $\boldsymbol{x}^{(2l)} \in \mathbb{R}^L$.

**$l$-th routing mechanism (Layer $2l - 1$ and gating in Layer $2l$).**

*Proof sketch:* These components work together to *exactly assign $x_l$ to its correct expert* $f^{(2l,i)}$. Specifically, the first MoE layer $h^{(2l-1)}$ behaves like a dense model, serving as an indicator function. Then the Layer-2 gating network $g^{(2l)}$ selects the expert corresponding to the region $U_i$ such that $x_l \in U_{l,i}$.

Consider the specific indicator function comprising 3 ReLU neurons:

$$\tau_{l,i}(x_l) := \text{ReLU}(x_l - (i-1)) + \text{ReLU}(x_l - i) - 2\text{ReLU}(x_l - (i - 1/2)), \quad i \in [E].$$

which satisfies:

$$\tau_{l,i}(x_l) \begin{cases} > 0 & \text{if } x_l \in (i-1, i) \\ 0 & \text{else} \end{cases}. \tag{13}$$

Let $\tau^{(l)}$ be the concatenated two-layer network of the $E$ two-layer networks, i.e., $\tau^{(l)}(\boldsymbol{x}_l) = (\tau_{l,i}(\boldsymbol{x}_l), \cdots, \tau_{l,E}(\boldsymbol{x}_l))^\top$. Note that the width of $\tau^{(l)}$ is only $3E$.

In Layer $2l - 1$, each expert network $f^{(2l-1,i)} : \mathbb{R}^L \to \mathbb{R}^{L+E}$ shares the same form:

$$f^{(2l-1,i)}(\boldsymbol{x}^{(2l-2)}) = \begin{pmatrix} \boldsymbol{x}^{(2l-2)} \\ \tau^{(l)}(x_l) \end{pmatrix}.$$

The weight matrix $\boldsymbol{W}_R^{(2l-1)}$ of the gating network $g^{(2l-1)} : \mathbb{R}^L \to \mathbb{R}^E$ can be any constant, such as $\boldsymbol{0}$. In fact, Layer $2l - 1$ behaves as a standard dense network computing $\tau^{(l)}(x_l)$.

For the gating network $g^{(2l)}$ in Layer $2l$, we define its weight matrix as:

$$\boldsymbol{W}_R^{(2l)} = \left(\boldsymbol{0}_{E \times L}, \boldsymbol{I}_{E \times E}\right) \in \mathbb{R}^{E \times (L+E)}.$$

Now we check the routing mechanism: given an input $\boldsymbol{x} = (x_1, \cdots, x_L) \in [0, E]^L$, where $x_l \in [0, E]$, Layer $2l - 1$ outputs

$$\boldsymbol{x}^{(2l-1)} = h^{(2l-1)}(\boldsymbol{x}^{(2l-2)}) = \begin{pmatrix} \boldsymbol{x}^{(2l-2)} \\ \tau^{(l)}(x_l) \end{pmatrix} \in \mathbb{R}^{L+E}.$$

Then the gating output of the Layer $2l$ is

$$g^{(2l)}(\boldsymbol{x}^{(2l-1)}) = \boldsymbol{W}_R^{(2l)} h^{(2l-1)}(x_l) = \tau^{(l)}(x_l) \in \mathbb{R}^E.$$

Then it will assign $x_l$ to the $i_{x_l}$-th expert network in Layer 2, where

$$i_{x_l} := \arg\max_{i \in [E]} g_i^{(2l)}(x_l) = \arg\max_{i \in [E]} \tau_{l,i}(x_l).$$

From the property (13), we have:

$$x_l \in U_{i_{x_l}}. \tag{14}$$

This establishes exact assignment of $x_l$ to its correct region.

### $l$-th expert networks (Layer $2l$).

*Proof sketch:* Each expert network $f^{(2l,i)}$ in Layer 2 is responsible for *approximating the 1-dimensional local target functions $f_{l,i}$*.

By Theorem 3.1, there exist two-layer dense networks $g_{l,i}$, with width $m$, such that

$$\|f_{l,i} - g_{l,i}\| \leqslant \tilde{\mathcal{O}}\left(\max\left\{m^{-\kappa(f_{l,i})}, m^{-1/2}\right\}\right) \leqslant \tilde{\mathcal{O}}\left(m^{-\kappa(f_{l,i}) \wedge \frac{1}{2}}\right).$$

Let the expert network $f^{(2l,i)} : \mathbb{R}^{L+E} \to \mathbb{R}^L$ as

$$f^{(2l,i)}(\boldsymbol{x}^{(2l-1)}) = \begin{pmatrix} \boldsymbol{x}_{1:l-1}^{(2l-1)} \\ g_{l,i}(x_l) \\ \boldsymbol{x}_{l+1:E}^{(2l-1)} \end{pmatrix},$$

which is a three-layer dense network and satisfies:

$$\sup_{x_l \in U_{l,i}} \left|f_{l,i}(x_l) - f_l^{(2l,i)}(\boldsymbol{x}^{(2l-1)})\right| = \sup_{x_l \in U_{l,i}} |f_{l,i}(x_l) - g_{l,i}(x_l)| \tag{15}$$
$$\leqslant \|f_{l,i} - g_{l,i}\| \leqslant \tilde{\mathcal{O}}\left(m^{-\kappa(f_{l,i} \circ \phi_{l,i}^{-1}) \wedge \frac{1}{2}}\right) = \tilde{\mathcal{O}}\left(m^{-\kappa(f_{l,i}) \wedge \frac{1}{2}}\right).$$

### The final estimate.

From the above construction, the output of the $2L$-th layer takes the form:

$$\Psi(\boldsymbol{x}) := \boldsymbol{x}^{(2L)} = \begin{pmatrix} g_{1,i_{x_1}}(x_1) \\ \vdots \\ g_{L,i_{x_L}}(x_L) \end{pmatrix} \in \mathbb{R}^L.$$

Now consider an input $\boldsymbol{x} = (x_1, \cdots, x_L) \in [0, E]^L$. For each $l \in [L]$, the $l$-th routing mechanism assigns it to the correct expert network $f^{(2l,i_{x_l})}$, which satisfies $x_l \in U_{i_{x_l}}$ (14). Consequently, the approximation error holds:

$$\|f(\boldsymbol{x}) - \Psi(\boldsymbol{x})\| = \max_{l \in [L]} \left|f_{l,i_{x_l}}(x_l) - g_{l,i_{x_l}}(x_l)\right|$$

$$\overset{(15)}{\leqslant} \max_{l \in [L]} \tilde{\mathcal{O}} \left( m^{-\kappa(f_{l,i_{x_l}}) \wedge \frac{1}{2}} \right) \leqslant \max_{l \in [L]} \max_{i \in [E]} \tilde{\mathcal{O}} \left( m^{-\kappa(f_{l,i}) \wedge \frac{1}{2}} \right).$$

Since $x$ is arbitrary, this concludes the proof.

## C  Experimental Details

The experiments in Section 6 are conducted on 1 A100 GPU.

In Experiment I, the models are trained for $2,000$ iterations with batch size $128$ (online), using squared loss and Adam optimizer with learning rate `1e-3`.

In Experiment II, the models are trained for $5,000$ iterations with batch size $128$ (online), using squared loss and Adam optimizer with learning rate `1e-3`.

