# OpenReview forum: "On the Expressive Power of Mixture-of-Experts for Structured Complex Tasks"
_NeurIPS.cc/2025/Conference — NeurIPS 2025 spotlight_

### Official Review · Reviewer_4U7j · 2025-06-12

**Clarity:** 4
**Significance:** 3
**Originality:** 3
**Rating:** 5
**Confidence:** 3

**Summary:**

The paper presents a theoretical analysis of expressivity of mixture-of-experts (MoE) networks in the context of complex tasks governed by two structural priors: low-dimensionality and sparsity.

**Questions:**

- What is the concrete definition of each MoE layer $h^{(l)}$, including the dimensionality of its inputs and outputs? Providing a clear and detailed definition would help readers understand the MoE model more easily.

- In practical applications, how large does the layer depth $L$ compare to the input dimension $d$ for a given target function in practice? Expanding on this point could help readers see how the theoretical result overcomes the curse of dimensionality.

- (minor, a typo?) Should the assumption in Theorem 5.3 be "Let the target function $f$ be of the form (5)" rather than "of the form (4)"?

**Ethical Concerns:**

["NO or VERY MINOR ethics concerns only"]

**Final Justification:**

After the rebuttal and discussions, I will keep my positive evaluation for the paper.

**Quality:**

3

**Strengths And Weaknesses:**

**Strengths**:
- The paper provides a solid theoretical framework with clear proofs outlining the approximation capabilities of both shallow and deep MoE networks.
- The work brings a novel angle to the study of network expressivity. This analysis bridges the gap between empirical successes of modern deep learning models and their theoretical understanding.
- The discussion on architectural design (e.g., nonlinear gating and alternating structures) is valuable for practitioners, as it provides insights into how theoretical improvements can translate into better network designs for real-world tasks.

**Weakness**: There is no major weakness in this work in my opinion. Some minor weaknesses are included in **Question** part.

---

> ### Author Rebuttal · Authors · 2025-07-30
>
> We sincerely thank the reviewer for the positive support and valuable feedback. We greatly appreciate the insightful review and the recognition of the significance of our contributions. The comments are highly constructive and will help us further improve our work. Below, we provide our point-by-point responses.
>
> ----------------------------------------
>
> > **Q1. Definition of MoE layer.** What is the concrete definition of each MoE layer $h^{(l)}$, including the dimensionality of its inputs and outputs? Providing a clear and detailed definition would help readers understand the MoE model more easily.
>
> **Response:** We thank the reviewer for the suggestion to clarify the definition. For simplicity, let us denote an MoE layer $h^{(l)}$ by $h$, which comprises a routing network $g$ and expert networks $f^{(k)},k\in[E]$. The definition of $h$ follows the procedure described in Lines L103-L105: given an input $\boldsymbol{x}\in\mathbb{R}^{d_{\rm in}}$, the output is $h(\boldsymbol{x}):=\boldsymbol{y}\in\mathbb{R}^{d_{\rm out}}$, where $\boldsymbol{y}$ is defined in L104. Here, $d_{\rm in},d_{\rm out}\in\mathbb{N}$ are general positive integers. In the revised version, we will include addiontional clarification to improve readability and completeness.
>
> ----------------------------------------
>
> > **Q2. Depth vs. dimensionality.** In practical applications, how large does the layer depth $L$ compare to the input dimension $d$ for a given target function in practice? Expanding on this point could help readers see how the theoretical result overcomes the curse of dimensionality.
>
> **Response:**
> We thank the reviewer for this insightful question. According to our theory in Section 5, the required depth $L$ depends not on the input dimension $d$, but on the compositional sparsity of the target function. In particular, Theorem 5.2 shows that for approximating functions of the compositional form (5), the required network depth is $2L$, independent of how large $d$ is. In comparison, our results in Section 4 emphasis that shallow MoE networks can overcome the curse of dimensionality when the target function is supported on low-dimenisonal manifolds.
>
>
> Additionally, to support our theoretical findings, we have conducted **two new experiments**:
> - **Experiment I. Shallow MoEs for low-dimensional functions.**
> - **Experiment II. Deep MoEs for piecewise functions.**
>
> To avoid redundancy, we kindly refer the reviewer to our **response to W1 from Reviewer 4bkA**,  where we describe the experimental setups and results in detail.
>
> ----------------------------------------
>
> > **Q3. Clarification on Theorem 5.3.** (minor, a typo?) Should the assumption in Theorem 5.3 be "Let the target function $f$ be of the form (5)" rather than "of the form (4)"?
>
> **Response:** We thank the reviewer for this question. We would like to clarify that form (4) is a special case of the more general compositional structure given in form (5). While Theorem 5.2 addresses the general case (5), Theorem 5.3 focuses on the special case of form (4), which allows us to derive sharper bounds on the required depth and width of the expert networks.

---

> > ### Comment · Reviewer_4U7j · 2025-08-03
> >
> > Thank you to the authors for the response and detailed clarifications. I have no further questions, and I would like to congratulate you on the good work.

---

> > > ### Author Response · Authors · 2025-08-04
> > >
> > > We would like to reiterate our sincere gratitude for your valuable recommendation and positive feedback. We appreciate your support very much!

---

### Official Review · Reviewer_9CGn · 2025-07-02

**Clarity:** 4
**Significance:** 4
**Originality:** 3
**Rating:** 4
**Confidence:** 3

**Summary:**

In this article, the authors illustrate the capacity of Mixture of Expert (MoE) model for approximating a function $f$ over a manifold $\mathcal{M}$. Based on the known result for two-layer ReLU networks (Theorem 3.1) in \cite{}, the main result (Theorem 4.8) shows that the MoE model can approximate with an error of $\mathcal{O}(m^{-(\kappa/d \wedge 1/2)})$, where $d$ is the intrinsic dimension, $\kappa$ control the smoothness of function $f$, and $m$ controls the width of network.
This result demonstrates the outstanding ability of the MoE model over shallow dense network, which approximate with error $\mathcal{O}(m^{(-\kappa/D\wedge 1/2)})$, where $D$ is the ambient dimension of $\mathcal{M}$. The author formularises the multiple tasks problem using piecewise functions and product of manifold, and derives the analogous result for this situation, which similarly shows the capacity of MoE to approximate a function with error depennds on intrinsic dimension.

**Questions:**

1. The article mentions the concept of reach, how can we use and how important this concept in this article?

2. In equation 7, there may be some situations that $x$ belongs to two atlas, so what happens in this case?

3.  As I mention in strength and weakness part, can we extend the model to the case of other activating function rather than ReLU (like softmax, GeLU, etc,...)

**Ethical Concerns:**

["NO or VERY MINOR ethics concerns only"]

**Final Justification:**

Thanks for the response and clarifications. I appreciate the effort you put into addressing these points in detail.
I will remain my score.

**Limitations:**

yes

**Quality:**

3

**Strengths And Weaknesses:**

Strengths:

1.  The result is nice overall and meaningful, which can significantly improve the known bound $\mathcal{O}(m^{-(\kappa/D \wedge 1/2)})$ to $\mathcal{O}(m^{-(\kappa/d \wedge 1/2)})$. This improvement shows the adaptability of MoE model in the case of low dimension.

2. The author provides a significant suggestion for MoE mechanism, such as non-linear gating function, alternation between MoE network and dense network.

3. The writing is clear and easy to read, the new concept is well-explained with several examples.

Weaknesses:

1. The paper should provide some experiments to support the result for the case of low dimension.

2. Besides the algorithm that prove the existence of good enough MoE, the paper should provide an algorithm to find a feasible neural network.

3. It appears that the model only consider one kind of gating function (ReLU), so it would be much better to consider other gating functions, like GeLU, softmax, etc.

Despite of some experimental drawback in the paper, the theoretical result here is excellent and interesting, which provides a rigorous justification for the adaptability of MoE model.

---

> ### Author Rebuttal · Authors · 2025-07-30
>
> We sincerely thank the reviewer for the positive support and valuable feedback. We greatly appreciate the insightful review and the recognition of the significance of our contributions. The comments are highly constructive and will help us further improve our work. Below, we provide our point-by-point responses.
>
> ----------------------------------------
>
> > **W1. Suggestion on experiments.** The paper should provide some experiments to support the result for the case of low dimension.
>
> **Response:** We thank the reviewer for the constructive suggestion to include empirical results, particularly regarding the low-dimensional case. To support our theoretical findings, we have conducted **two new experiments**:
> - **Experiment I. Shallow MoEs for low-dimensional functions.**
> - **Experiment II. Deep MoEs for piecewise functions.**
>
> To avoid redundancy, we kindly refer the reviewer to our **response to W1 from Reviewer 4bkA**,  where we describe the experimental setups and results in detail.
>
> ----------------------------------------
>
> > **W2. Training algorithms.** Besides the algorithm that prove the existence of good enough MoE, the paper should provide an algorithm to find a feasible neural network.
>
> **Response:** We thank the reviewer for this suggestion. Our current results focus on the expressivity of MoE archetuctues, which serves as a foundational step toward a deeper theoretical understanding of MoEs. As noted at Section 6, a key direction for future work is to investigate whether such expressive solutions can be discovered through optimization algorithms. Analyzing this question presents additional challenges, including MoE-specific training instability and the highly non-convex loss landscape. We leave this important direction for future investigation.
>
> ----------------------------------------
>
> > **W3 & Q3. Extension to other activation functions.** It appears that the model only consider one kind of gating function (ReLU), so it would be much better to consider other gating functions, like GeLU, softmax, etc. As I mention in strength and weakness part, can we extend the model to the case of other activating function rather than ReLU (like softmax, GeLU, etc,...)
>
> **Response:**
> We thank the reviewer for this constructive question. Although our current analysis is based on ReLU-activated experts, the theoretical framework extends naturally to MoEs with other activation functions (e.g., GeLU, SiLU, Swish). This extension is feasible because our results rely on prior study establishing the expressivity of two-layer ReLU networks (Theorem 3.1). To extend the results, we only need to substitute the ReLU-specific approximation results with analogous results for the desired activation functions. For instance, for GeLU, SiLU, and Swish, the approximation theory developed in [1] can be used. In the revised version, we will include a detailed discussion of this extension.
>
> ----------------------------------------
>
> > **Q1. About the concept of reach.** The article mentions the concept of reach, how can we use and how important this concept in this article?
>
> **Response:** We thank the reviewer for this thoughtful question. The concept of reach is introduced in the appendix to illustrate that common low-dimensional manifolds with positive reach admit well-structured atlas, where the chart maps are linear. For example, *Example 4.6 satisfies this condition*. We apologize that this connection is not clearly stated in the current version. In the revised version, we will explicitly mention this it in Example 4.6 and elaborate further in the appendix.
>
> ----------------------------------------
>
> > **Q2. Overlapping atlas regions.** In equation 7, there may be some situations that $x$ belongs to two atlas, so what happens in this case?
>
> **Response:** We thank the reviewer for this insightful question. When an input $x$ ies in the overlap of two atlas regions, $U_i$ and $U_j$, both corresponding experts, $f^{(i)}$ and $f^{(j)}$, can provide efficient approximation over the intersection $U_i\cap U_j$. In this case, the router may assign $x$ to either expert $i$ or $j$; i.e., the index $i_{\boldsymbol{x}}$ in Eq. (7) can be either $i$ or $j$. Regardless of which expert is selected, the resulting approximation remains valid and accurate.
>
> ----------------------------------------
>
> **References**
>
> [1] Zhang et al., Deep Network Approximation: Beyond ReLU to Diverse Activation Functions. Journal of Machine Learning Research, 2024.

---

> > ### Comment · Reviewer_9CGn · 2025-08-05
> >
> > Thanks for the response and clarifications. I appreciate the effort you put into addressing these points in detail.

---

> > > ### Author Response · Authors · 2025-08-05
> > >
> > > We would like to reiterate our sincere gratitude for your valuable recommendation and positive feedback. We appreciate your support very much!

---

### Official Review · Reviewer_rAT9 · 2025-07-03

**Clarity:** 3
**Significance:** 2
**Originality:** 3
**Rating:** 4
**Confidence:** 2

**Summary:**

The authors analyze the expressive power of sparse MoE models when data (1) lies on a low-dimensional manifold or (2) exhibits compositional sparsity. They find that sparse MoEs can match the approximation power of comparable dense networks while evaluating only $1/E$ of the expert parameters per input in both the (1) low-dimensional manifold setting and (2) the compositional sparsity setting, ignoring the routing parameters and assuming the router can correctly route to the correct expert.

**Questions:**

* Is there potentially an issue with how a Top-1 gate maintains a clear margin so that each input is routed to a single expert, especially near chart boundaries?
* In the deep MoE setting, have you looked at how small routing errors in one layer might propagate through the stack, and whether a bound on that accumulation is possible?

**Ethical Concerns:**

["NO or VERY MINOR ethics concerns only"]

**Final Justification:**

The paper appears to be theoretically solid. The authors responded to my questions well, and the paper seems well-regarded by the other reviewers. I'll keep my confidence relatively low at 2.

**Limitations:**

yes

**Quality:**

3

**Strengths And Weaknesses:**

### Strengths
* The paper is clearly written and presented
* Results seem interesting and plausible, though I have not carefully verified the results

### Weaknesses
* On line 319, should the expression currently written as
$$ ( f_{1,i_1}(x_1),\cdots, f_{1,i_L}(x_L) ) $$
instead be:
$$ ( f_{1,i_1}(x_1),\cdots, f_{L,i_L}(x_L) ) $$
* The paper's claims may hinge on assumptions about the router, in particular that it picks the one correct expert for every input.
* The paper contains no empirical comparisons or discussions of MoE scaling laws, although this is standard for expressivity papers.

---

> ### Author Rebuttal · Authors · 2025-07-30
>
> We sincerely thank the reviewer for the positive support and valuable feedback. We greatly appreciate the insightful review and the recognition of the significance of our contributions. The comments are highly constructive and will help us further improve our work. Below, we provide our point-by-point responses.
>
> ----------------------------------------
>
> > **W1. A typo.** On line 319, should the expression currently written as $(f_{1,i_1}(x_1),\cdots,f_{1,i_L}(x_L))$ instead be: $(f_{1,i_1}(x_1),\cdots,f_{L,i_L}(x_L))$.
>
> **Response:** We are grateful to the reviewer for identifying the typo. In the revised version, we will carefully read through the whole paper and correct all typos.
>
> ----------------------------------------
>
> > **W2. Assumptions on the router.** The paper's claims may hinge on assumptions about the router, in particular that it picks the one correct expert for every input.
>
> **Response:** We thank the reviewer for raising this insightful point. We will incorporate a discussion of this issue in the revised version. Our response is as follows:
>
> - First, our results are constructive and concern the expressive power of MoE networks. We explicitly prove the existence of a MoE which can **perfectly assign** each input to its correct expert. However, as discussed in Section 6, a key open question is whether such solutions can be discovered by training algorithms such as stochastic gradient descent.
>
> - **Extension to Top-$K$ routing.** Our results extend naturally from top-1 routing to top-$K$ routing. For example, given an MoE network $\Psi$ using top-$1$ routing, we can construct another MoE network $\Psi'$ using top-$2$ routing, with twice as many experts by duplicating each expert in expert in $\Psi$. Notably, $\Psi'$ has equivalent expressivity as $\Psi$.
>
> - **Efficiency of Top-1 routing.** In some cases, top-1 routing may be nearly optimal. For example, under the setting of Theorem 4.8, some input $x$ belongs exclusively to a single subregion $U_i$, and activating additional experts may actually increase approximation error. This view aligns with recent empirical findings suggesting that increased sparsity can improve MoE performance (e.g., Kimi-K2 technical report, 2025.07).
>
> ----------------------------------------
>
> > **W3. Suggestion on experiments.** The paper contains no empirical comparisons or discussions of MoE scaling laws, although this is standard for expressivity papers.
>
> **Response:** We thank the reviewer for the valuable suggestion to include empirical results. To support our theoretical findings, we have conducted **two new experiments**:
> - **Experiment I. Shallow MoEs for low-dimensional functions.**
> - **Experiment II. Deep MoEs for piecewise functions.**
>
> To avoid redundancy, we kindly refer the reviewer to our **response to W1 from Reviewer 4bkA**,  where we describe the experimental setups and results in detail.
>
> ----------------------------------------
>
> > **Q1. Routing near chart boundaries.** Is there potentially an issue with how a Top-1 gate maintains a clear margin so that each input is routed to a single expert, especially near chart boundaries?
>
> **Response:** We thank the reviewer for this interesting question.
> When an input $x$ lies in the overlap of two atlas regions, $U_i$ and $U_j$, (e.g., near the chart boundaries), both corresponding experts, $f^{(i)}$ and $f^{(j)}$, can provide efficient approximation over the intersection $U_i\cap U_j$. By our construction (detailed in the Proof of Theorem 4.8), the router may assign $x$ to either expert $i$ or $j$. Regardless of which expert is selected, the resulting approximation remains valid and accurate.
>
> ----------------------------------------
>
> > **Q2. Propagation of routing errors in deep models.** In the deep MoE setting, have you looked at how small routing errors in one layer might propagate through the stack, and whether a bound on that accumulation is possible?
>
> **Response:** We thank the reviewer for this insightful question.
> - **Theoretical expressivity.** Our main theorems establish the existence of a router that **perfectly assigns** each input to its correct expert. In this idealized setting, routing errors do not occur, and hence no error propagation arises.
>
> - **Practical training.** During training, however, routing errors are inevitable. We hypothesize that in the worst case, such errors may propagate multiplicatively with depth. Nevertheless, if MoEs have learned good representations, e.g., if each layer specializes in a correct class of subtasks, as in Equation (5), then routing errors may not accumulate significantly but instead remain stable across layers. This would be an exciting direction for future understanding.

---

> > ### Comment · Reviewer_rAT9 · 2025-08-03
> >
> > Thank you for your comments addressing my questions. In light of this response and the comments from other reviewers, I'll likely increase my score to a 5.

---

> > > ### Author Response · Authors · 2025-08-04
> > >
> > > We would like to reiterate our sincere gratitude for your valuable recommendation and positive feedback. We are pleased to have addressed your main concern and appreciate your decision to raise the score. Thank you!

---

### Official Review · Reviewer_4aPH · 2025-07-03

**Clarity:** 4
**Significance:** 4
**Originality:** 3
**Rating:** 5
**Confidence:** 3

**Summary:**

This paper draws the parallel between MoE networks with structured tasks, by showing an MoE model having representation capacity of a $E^L$-piece piecewise function class.

**Questions:**

See Weaknesses (for lack of better naming).

**Ethical Concerns:**

["NO or VERY MINOR ethics concerns only"]

**Final Justification:**

I still think this is a solid Accept. It isn't groundbreaking enough to get a 6, and very much deserves a 5.

**Limitations:**

Paper listed potential mismatch between theory and real-world problems, and whether SGD can train such a (theoretical) MoE model. No potential impact is listed.

**Paper Formatting Concerns:**

No paper formatting concerns.

**Quality:**

4

**Strengths And Weaknesses:**

### Strengths:
- Introduction, motivation, and organization is clear; with annotation in theorems/equations whenever available.
- Rigorous proof for a simple-sounding concept, specifically analyzing the un-simplified (i.e. close to the real implementation) linear MoE router.

### Weaknesses
- Difficult to follow notations.
  - Overloaded notation: $\alpha$ were used as softmax outputs (L105), then an integer permutation vector (L117)
  - Related, what is $\mathcal{A}$?
  - Similarly, $f|_{U_i}$ in Theorem 4.8 onwards were never formally defined.
- Albeit this is a purely theoretical work, experimental results would greatly benefit the persuasion of the paper.
  - The authors could _manually_ craft a theoretical 4-3-MoE to fit the toy problem in Fig. 3, then compare it to baselines such as VI.

---

> ### Author Rebuttal · Authors · 2025-07-30
>
> We sincerely thank the reviewer for the positive support and valuable feedback. We greatly appreciate the insightful review and the recognition of the significance of our contributions. The comments are highly constructive and will help us further improve our work. Below, we provide our point-by-point responses.
>
> ----------------------------------------
>
> > **W1. Ambiguities in notation.** "Difficult to follow notations. Overloaded notation: $\alpha$ were used as softmax outputs (L105), then an integer permutation vector (L117); Related, what is $\mathcal{A}$?; Similarly, $f|_{U_i}$ in Theorem 4.8 onwards were never formally defined."
>
> **Response:** We thank the reviewer for carefully pointing out these ambiguities. In the revised version, we will thoroughly review the notation to ensure consistency and clarity. Specifically,
> - The symbol $\alpha$ is indeed overloaded. We will revise its usage at L117 to $\beta$.
> - The symbol $\mathcal{A}$ refers to an index set. In most cases in this work, it is simply $\mathcal{A}=[E]$, where $E$ is a positive integer.
> - The notation $f|\_{U_i}$ refers to the restriction of a function $f:\mathcal{M}\to\mathbb{R}$ to a subset $U_i\subset\mathcal{M}$. That is, $f|\_{U_i}:U_i\to\mathbb{R}$ is defined by $f|\_{U_i}(\boldsymbol{x}):=f(\boldsymbol{x})$ for all $\boldsymbol{x}\in U_i$.
>
> ----------------------------------------
>
> > **W2. Suggestion on experiments.** "Albeit this is a purely theoretical work, experimental results would greatly benefit the persuasion of the paper. The authors could manually craft a theoretical 4-3-MoE to fit the toy problem in Fig. 3, then compare it to baselines such as VI."
>
> **Response:** We thank the reviewer for the constructive suggestion to include empirical results, particularly regarding fitting the toy problem in Figure 3. To support our theoretical findings, we have conducted **two new experiments**:
> - **Experiment I. Shallow MoEs for low-dimensional functions.**
> - **Experiment II. Deep MoEs for piecewise functions.**
>
> To avoid redundancy, we kindly refer the reviewer to our **response to W1 from Reviewer 4bkA**,  where we describe the experimental setups and results in detail.
>
> Specifically, our Experiment II demonstrates that a 2-3-MoE can successfully fit the function in Figure 3. While our theoretical analysis considers a more general case and suggests that 4 layers may be required, we found that 2 layers are both sufficient and necessary for this specific example.
>
> Lastly, we kindly ask the reviewer to clarify what is meant by "baselines such as VI." We would be happy to consider it in future empirical comparisons.

---

> > ### Comment · Reviewer_4aPH · 2025-08-05
> >
> > Thank you for the clarification. When referring to VI, I was alluding to modelling a piecewise function with a Gaussian mixture model, which is a natural approach. The provided additional experiments are actually quite amazing, although the conditions are still very tame, with a small number of pieces in a small region. One thing I would like to note is that the evaluated losses are closely reaching the default precision limit, so the authors might want to keep that in mind. I hope the authors address the two new concerns I have in the final revision of the paper. Overall, congratulations on the great work!

---

> ### Author Response · Authors · 2025-08-05
>
> Thank you for clarifying the "VI". We appreciate your insightful suggestion to experiment with modeling a piecewise function using a Gaussian mixture model. We will incorporate this experiment in the final revision.
>
> We also thank you for pointing out the potential precision-related issue. We would like to clarify that, since we use full-precision training, the evaluated losses in our setup have not yet reached the default precision limit; they can indeed go below the order of 1e-10. Nonetheless, we will ensure that the precision level is appropriately monitored and reported in the final version.
>
> Overall, we would like to reiterate our sincere gratitude for your valuable recommendation and positive feedback. We appreciate your support very much!

---

### Official Review · Reviewer_4bkA · 2025-07-05

**Clarity:** 3
**Significance:** 3
**Originality:** 3
**Rating:** 5
**Confidence:** 3

**Summary:**

The paper conducts a systematic study on the theoretical foundations of Mixture-of-Experts (MoE) networks in modeling complex tasks. It demonstrates that shallow MoE networks can efficiently approximate functions supported on low-dimensional manifolds, overcoming the curse of dimensionality. For deep MoE networks, the paper shows that they can approximate piecewise functions with compositional sparsity, enabling the modeling of an exponential number of structured tasks. The analysis also provides insights into the roles of architectural components and hyperparameters in MoEs and offers practical suggestions for MoE variants.

**Questions:**

See my comments above.

**Ethical Concerns:**

["NO or VERY MINOR ethics concerns only"]

**Final Justification:**

Thanks for providing more experiments and discussing the influence of token routing strategies. The paper would be greatly improved with it. Therefore, I am happy to raise my score to 5. Thanks for the efforts to address my previous comments :)

**Quality:**

3

**Strengths And Weaknesses:**

**Strengths**:
(1) The paper provides a solid theoretical analysis of the expressive power of both shallow and deep MoE networks, filling a gap in the understanding of their capabilities, thus a valuable contribution to the community.

(2) It offers practical suggestions for designing MoE architectures, such as incorporating nonlinearity into gating mechanisms and using low-dimensional expert networks via autoencoding.

(3) The results apply to a wide range of structured tasks, demonstrating the versatility and potential of MoE networks in various domains.

(4) The paper effectively compares the efficiency of MoE networks with dense networks, highlighting the computational advantages of MoEs.


**Weaknesses**:
(1) I understand this paper is more of a theory paper. However, it would be great if the authors can design some synthetic tasks (e.g., low dimensional or piecewise functions) to better illustrate their theoretical results. It would strengthen the claims and provide practical evidence of the proposed methods.

(2) The analysis focuses primarily on compositional sparsity, and other forms of sparsity such as temporal sparsity on the sequential data are not explored, which might be related to the real-world language understanding task where we have seen MoE's successful usage.

(3) Some recent studies have shown that token routing strategy has significant effects on the MoE performance (see references 1-4 from the list below). Just wondering whether the theoretical results in this paper would be affected by using different routing strategies in any way. It would be great if the authors can include some discussions about it.

**Reference**:
[1] Nguyen, et al. "Statistical Advantages of Perturbing Cosine Router in Sparse Mixture of Experts." arXiv preprint arXiv:2405.14131 (2024).
[2] Liu, et al. "Gating dropout: Communication-efficient regularization for sparsely activated transformers." International Conference on Machine Learning. PMLR, 2022.
[3] Zuo, et al. "Taming sparsely activated transformer with stochastic experts." arXiv preprint arXiv:2110.04260 (2021).
[4] Lewis, et al. "Base layers: Simplifying training of large, sparse models." International Conference on Machine Learning. PMLR, 2021.

---

> ### Author Rebuttal · Authors · 2025-07-30
>
> We sincerely thank the reviewer for the positive support and valuable feedback. We greatly appreciate the insightful review and the recognition of the significance of our contributions. The comments are highly constructive and will help us further improve our work. Below, we provide our point-by-point responses.
>
> ----------------------------------------
>
> > **W1. Suggestion on experiments.** "I understand this paper is more of a theory paper. However, it would be great if the authors can design some synthetic tasks (e.g., low dimensional or piecewise functions) to better illustrate their theoretical results. It would strengthen the claims and provide practical evidence of the proposed methods."
>
> **Response:**
> Thank the reviewer for offering this constructive suggestion. To support our main theoretical results, we have conducted **two new experiments**, each aligned with one of our key insights.
>
> - **Experiment I. Shallow MoEs for low-dimensional functions.**
>   - **Objective.** To validate our theoretical insight in Section 4 (Theorem 4.8): shallow MoE networks can efficiently approximate functions supported on low-dimensional manifolds and *overcome the curse of dimensionality*.
>
>   - **Setup.**
>     - *Target Function*. Consider the low-dimensional manifold $\mathcal{M}$={$\boldsymbol{x}\in\mathbb{R}^D:x_1^2+x_2^2=1;x_i=0,\forall i>2$} embedded in $\mathbb{R}^D$ with $D>2$. The target function is $f(\boldsymbol{x})=\sin(5x_1)+\cos(3x_2)$, defined on $\mathcal{M}$.
>     - *1-4-MoE model*. We consider a 1-layer MoE comprising 1 router and 4 experts. Each expert is a two-layer ReLU network with hidden width $10$.
>     - *Training Algorithm*. To validate whether MoE can overcome the curse of dimensionality, we vary the input dimension $D\in${16,32,64,128}. Models are trained for 2,000 iterations with batch size 128 (online), using squared loss and Adam optimizer with learning rate 1e-3.
>
>
>   - **Results & Conclusion.** The following table shows the test error of 1-4-MoE under different input dimensions $D$.
>
>     | input dim $D$ | $16$ | $32$ | $64$ | $128$ |
>     | :----: | :----: | :----: | :----: | :----: |
>     | **test error** | 3.40e-4 | 3.38e-4 | 3.17e-4 | 3.42e-4 |
>
>     As $D$ increases, the test error of MoE does not increase significantly and remains stable. This supports our insight that shallow MoEs efficiently approximate functions on low-dimensional manifolds and avoid the curse of dimensionality.
>
>
> - **Experiment II. Deep MoEs for piecewise functions.**
>
>   - **Objective.** To verify our theoretical insight in Section 5: depth-$O(L)$ MoE networks with $E$ experts per layers can efficiently approximate piecewise functions with $E^L$ distinct pieces.
>
>     - **Setup.**
>       - *Target Function*. As defined in our Figure 3, we consider the piecewise function $f$ with compositional sparsity defined over $3^2 = 9$ unit cubes.
>       - *2-3-MoE and 1-6-MoE models*. We consider a 2-layer MoE comprising 2 routing layers and 2 expert layers with 3 experts each. To illustrate the role of depth, we also consider a shallow 1-6-MoE, with comparable parameter count. Each expert is a two-layer ReLU FFN with hidden width $m\in${16,32,64,128}.
>       - *Training Algorithm*. To validate whether 2-3-MoE and 1-6-MoE can approximate this target, we vary the hidden width $m$. Models are trained for 5,000 iterations with batch size 128 (online), using squared loss and Adam optimizer with learning rate 1e-3.
>     - **Results & Conclusion.** The following table shows the test error of the 2-3-MoE and 1-6-MoE under different hidden width $m$.
>
>         | hidden width $m$ of experts | $16$ | $32$ | $64$ | $128$ |
>         | :----: | :----: | :----: | :----: | :----: |
>         | **test error of 2-3-MoE** | 8.32e-5 | 1.41e-5 | 4.73e-6 | 2.59e-6 |
>         | **test error of 1-6-MoE** | 7.96e-5 | 2.17e-5 | 2.65e-5 | 4.60e-5 |
>
>         One can see that:
>         - As $m$ increases, 2-3-MoE achieves rapidly decreasing error. This supports that the depth-$2$ MoE with $3$ experts per layers can efficiently approximate this piecewise function with $3^2$ distinct pieces.
>         - In contrast, 1-6-MoE exhibits a performance plateau, revealing its limited expressive power. This highlights the crucial role of depth in modeling such compositional structures.
>
> We will incorporate these results and figures in the revised version and add additional synthetic tasks to further illustrate our theoretical findings.
>
> ----------------------------------------
>
> > **W2. Other forms of sparsity.** The analysis focuses primarily on compositional sparsity, and other forms of sparsity such as temporal sparsity on the sequential data are not explored, which might be related to the real-world language understanding task where we have seen MoE's successful usage.
>
> **Response:** We thank the reviewer for this insightful suggestion. We agree that temporal sparsity is highly relevant to natual language tasks and deserves theoretical exploration. While the current analysis focuses on compositional sparsity, we believe the unified insight remains applicable: MoEs can effectively discover the underlying structure priors (temporal sparsity), and subsequently decompose them into simpler subproblems, each solved by specialized experts. We will explore this direction in future work.
>
> ----------------------------------------
>
> > **W3. Influence of routing strategies.** Some recent studies have shown that token routing strategy has significant effects on the MoE performance (see references 1-4 from the list below). Just wondering whether the theoretical results in this paper would be affected by using different routing strategies in any way. It would be great if the authors can include some discussions about it.
>
> **Response:** We thank the reviewer for raising this insightful point and for providing relevant references. In our revised version, we will expand the discussion near L213–L220. Below, we summarize our preliminary views:
>
> - **Linear routing.** As noted in L213–L220, since standard gating function is linear and lacks the capacity to model nonlinear partition functions, an additional MoE (or dense) layer is needed prior to gating to approximate partition functions.
> - **Highly nonlinear routing.** If the router incorporates sufficient nonlinearity, e.g., a two-layer ReLU routing network, it can directly model complex partitions, reducing the number of parameters required in experts. For instance, in Theorem 4.8, the required depth will reduce from $2$ to $1$, because the nonlinearity in router eliminates the need for a preceding MoE layer. Similarly, in Theorem 5.2, the required depth will reduce from $2L$ to $L$. We will formalize these results in the revised version.
> - **Other routing mechanisms.** Routing functions such as cosine routing [1] and quadratic routing [5] can also introduce nonlinearity, and may similarly reduce the parameter complexity of the experts. Additionally, routing dropout [2] and stochatic routing [3], though not more expressive than linear routing, may improve training stability and load balancing.
>
> ----------------------------------------
>
> **References**
>
> [5] Akbarian et al., Quadratic gating functions in mixture of experts: A statistical insight. arXiv preprint arXiv:2410.11222 (2024).

---

### Note · Authors · 2025-08-14

We would like to take this opportunity to thank the reviewers and the area chair for their efforts in evaluating our work and for their insightful, motivating comments, which have helped improve the paper.

We are grateful that **all reviewers** recognized the **novelty, strength, and breadth** of our contributions in establishing the expressive power of MoE networks for modeling complex tasks with two common structural priors. In particular, we have theoretically demonstrated that:

- **Shallow MoE** networks can efficiently approximate functions supported on low-dimensional manifolds, overcoming the curse of dimensionality.
- **Deep MoE** networks can approximate piecewise functions with compositional sparsity, enabling the modeling of an exponential number of structured tasks.
- Our analysis also provides **insights** into the roles of architectural components in MoEs and offers natural suggestions for MoE variants.

Following the rebuttal discussions, in the final version we will:

- Incorporate the complete settings and results of the two new **experiments** introduced during the rebuttal (detailed in our response to W1 from Reviewer 4bkA), i.e., Experiement I: Shallow MoEs for low-dimensional functions; Experiement II: Deep MoEs for piecewise functions. We will also add further experiments on synthetic tasks, such as modeling piecewise functions with Gaussian mixture models.
- Integrate the **discussion** with the reviewers, including the extension to other activation functions, the influence of routing strategies, a more detailed characterization of the reach, etc.
- Carefully review all notations and correct typos.


We hope our results can contribute to a more fundamental understanding of why MoE networks are so successful in modeling structured complex tasks in modern deep learning.

---

### Decision · Program_Chairs · 2025-09-17

**Decision:**

Accept (spotlight)

**Comment:**

This paper analyses the expressive power of mixture of experts (MoE).

In particular authors prove that  shallow MoE networks  approximate functions supported on low-dimensional manifolds, overcoming the curse of dimensionality. For deep MoE networks, the paper shows that they can approximate piecewise functions with compositional sparsity.

Reviewers were all appreciative of the paper and the analysis and they suggested adding illustrative experimental results which we are added during the rebuttal. Reviewers suggested discussing other form of sparsity such as the temporal one as well as different routing strategies or different non-linearities, authors provided elements of responses to this which should be incorporated in the final manuscript. In addition to this reviewers pointed to several notational issues that made some parts difficult to follow, authors promised fixing these.